# From Assumptions to Actions: Turning LLM Reasoning into Uncertainty-Aware Planning for Embodied Agents

**SeungWon Seo**[*], **SooBin Lim**[*], **Seongrae Noh**[*], **Haneul Kim**, **HyeongYeop Kang**[†]
Department of Computer Science and Engineering, Korea University
{ssw03270,dnpcs,rhosunr99,adsky0309,siamiz_hkang}@korea.ac.kr

## Abstract

Embodied agents operating in multi-agent, partially observable, and decentralized environments must plan and act despite pervasive uncertainty about hidden objects and collaborators' intentions. Recent advances in applying Large Language Models (LLMs) to embodied agents have addressed many long-standing challenges, such as high-level goal decomposition and online adaptation. Yet, uncertainty is still primarily mitigated through frequent inter-agent communication. This incurs substantial token and time costs, and can disrupt established workflows, when human partners are involved. We introduce PCE, a Planner-Composer-Evaluator framework that converts the fragmented assumptions latent in LLM reasoning traces into a structured decision tree. Internal nodes encode environment assumptions and leaves map to actions; each path is then scored by scenario likelihood, goal-directed gain, and execution cost to guide rational action selection without heavy communication. Across two challenging multi-agent benchmarks (C-WAH and TDW-MAT) and three diverse LLM backbones, PCE consistently outperforms communication-centric baselines in success rate and task efficiency while showing comparable token usage. Ablation results indicate that the performance gains obtained by scaling model capacity or reasoning depth persist even when PCE is applied, while PCE consistently raises the baseline across both capacity and reasoning-depth scales, confirming that structured uncertainty handling complements both forms of scaling. A user study further demonstrates that PCE produces communication patterns that human partners perceive as more efficient and trustworthy. Together, these results establish a principled route for turning latent LLM assumptions into reliable strategies for uncertainty-aware planning.

## 1 Introduction

Imagine two household agents collaborating to prepare a meal: each perceives only part of the kitchen, yet both must coordinate to assemble ingredients and deliver them to a shared workspace. Such agents exemplify embodied agents, which perceive their surroundings, formulate plans, and execute actions to achieve specified goals in dynamic environments (Fung et al., 2025). When multiple embodied agents must cooperate in a decentralized setting under partial observability, each agent's limited perceptual field produces pervasive uncertainties about unobserved objects as well as the intentions and actions of collaborators (Spaan et al., 2006; Amato et al., 2016; 2015). Naively exploring these uncertainties at every planning step is computationally prohibitive. It also tends to produce suboptimal or inconsistent plans, as the combinatorial growth of possible hidden states significantly increases the difficulty of assessing the relative value of available actions and maintaining a coherent belief.

Agents powered by Large Language Models (LLMs) have achieved notable progress in such settings. By leveraging few-shot and zero-shot reasoning, LLM-based planners can decompose high-level goals into executable action sequences, adapt plans to environmental changes, and reliably infer the intentions of the collaborators, enabling robust performance in complex long-horizon settings. To handle uncertainty, however, most existing systems rely on repeated natural-language

---

[*]These authors contributed equally to this work.
[†]Corresponding author.

communication with collaborators to verify plans, exchange information, and iteratively refine joint strategies. This communication-centric paradigm incurs significant costs: frequent dialogue consumes large numbers of tokens and time, and when human collaborators are involved, continuous questioning and reporting can disrupt established workflows. Moreover, simply increasing LLM capacity or deepening its reasoning chains does not inherently resolve uncertainty. Without explicit mechanisms to identify and assess the assumptions that underlie uncertainty in partially observable environments, larger models still struggle to weigh competing hypotheses about the environment, which leads to misaligned priorities and degraded planning quality.

To address these limitations, we leverage two key empirical observations: First, when asked to evaluate or select actions, LLM planners internally generate implicit assumptions about uncertain aspects of the environment in their zero-shot Chain-of-Thought reasoning traces. For example, they tend to hypothesize the presence of an unseen object or infer the likely action of a collaborator. These assumptions can in turn be used to identify what information is missing. Second, we found that such assumptions are invoked locally and referenced implicitly, without being explicitly aggregated for a global decision. This unstructured handling prevents the planner from systematically reconciling multiple assumptions, which hampers its ability to detect logical conflicts, compare expected gains, weigh alternative actions consistently, and ultimately achieve higher planning accuracy.

We therefore propose Planner-Composer-Evaluator (PCE) to extract and aggregate these internally generated assumptions into a single structured representation in the form of a decision tree, enabling them to be jointly evaluated for more reliable and uncertainty-aware action selection. In this tree, each internal node specifies an assumption about the environment, and each leaf node specifies the action considered appropriate under the accumulated assumptions along the path. Thus, each root-to-leaf path represents a particular combination of assumptions culminating in an action trajectory. We then introduce an evaluator that scores each path in terms of its likelihood, gain, and cost, guiding rational action selection without heavy communication. This design transforms the latent knowledge already produced by LLM reasoning into a principled mechanism for uncertainty-aware planning.

Unlike prior multi-agent planners (Seo et al., 2025; Liu et al., 2025; Zu et al., 2025) that primarily optimize coordination through iterative communication or joint plan negotiation, PCE operates at a different level of the decision problem. Instead of optimizing over action sequences or communication strategies, PCE explicitly treats environmental assumptions as first-class decision variables and reasons over them before action execution. This shifts the planning paradigm from communication-centric coordination to structured reasoning over uncertainty embedded in the agent's own belief state.

Experiments on two challenging multi-agent benchmarks, C-WAH (Zhang et al., 2024b) and TDW-MAT (Zhang et al., 2024b), show that PCE consistently outperforms communication-centric baselines in success rate, task efficiency, and token usage. Ablation studies reveal that increasing LLM capacity or its reasoning depth without explicit uncertainty handling yields only limited performance improvements relative to its computational cost. User studies further demonstrate improved reliability in human-agent collaboration. Because the proposed framework operates on generic reasoning traces rather than model-specific internals, it can be readily applied to a wide range of LLM backbones and multi-agent domains. We demonstrate this generality by running PCE on diverse backbones, including GPT-4o mini, GPT-OSS:20B, and Gemma3:4B, and observe consistent improvements across all of them. Together, these results shift the focus from communication-heavy mitigation to principled exploitation of the assumptions already present in LLM reasoning, opening a new direction for uncertainty-aware planning in embodied systems. Code and project page are available at `https://ssw03270.github.io/PCE_ICLR-26_Page/`.

## 2 RELATED WORK

**Embodied Multi-Agent Cooperation.** Embodied agents aim to achieve long-term goals in complex environments populated with diverse objects (Song et al., 2023; Li et al., 2024). Recent work has leveraged LLMs as planners to decompose high-level goals into executable action sequences for navigation and manipulation tasks (Zhou et al., 2024; Huang et al., 2022; Song et al., 2023; Wang et al., 2023a;c; Hwang et al., 2025) and to enhance planning effectiveness through zero-shot and few-shot reasoning as well as dynamic replanning (Huang et al., 2022; Song et al., 2023). Beyond

single-agent settings, cooperative planning in multi-agent environments has been studied to address tasks that exceed the capacity of an individual agent (Zhang et al., 2024c; Guo et al., 2024).

To study more realistic embodied multi-agent cooperation, researchers focus on environments where distributed agents must make decisions based on their own limited observations (Zhang et al., 2024b; Seo et al., 2025; Liu et al., 2025; Zu et al., 2025). In these settings, agents cannot acquire information beyond their observation range in real time. They must therefore either move physically to gather information or share it through communication with collaborators to reduce travel time (Oliehoek & Spaan, 2012; Spaan et al., 2006; Foerster et al., 2016).

Prior work has mitigated uncertainty through intensive communication. For example, ProAgent (Zhang et al., 2024a) explicitly reasons about collaborators' actions and verifies its reasoning through dialogue; CoELA (Zhang et al., 2024b) and REVECA (Seo et al., 2025) exchange state and plan information to validate and adjust strategies; RoCo (Mandi et al., 2024) and CaPo (Liu et al., 2025) introduce iterative inter-agent debates; and CoTS (Zu et al., 2025) exploits communication-based feedback to refine joint plans. While these approaches improve coordination, they incur high token and time costs and can disrupt workflows when humans are in the loop.

More recently, LLaMAR (Nayak et al., 2024) has introduced a long-horizon LLM-based planning framework that operates in partially observable environments without relying on heavy communication, yet its formulation addresses coordination in a centralized multi-agent setting.

Our work takes a different path: instead of relying on heavy communication, we extract and aggregate the implicit assumptions in LLM reasoning into a decision tree framework that jointly evaluates them for rational action selection. This enables agents to adaptively balance physical and communicative actions, reducing communication overhead while preserving or improving planning performance.

**Scaling LLM for Embodied Agents.** Recent advances in LLMs have enabled natural language reasoning and high-level decision making for embodied agents. Scaling model capacity improves performance (Kaplan et al., 2020) but requires substantial training time and data resources. To complement these costs, reinforcement learning fine-tuning has been proposed at the training stage (Shao et al., 2024; Wang et al., 2024; Xu et al., 2025), while reasoning-augmentation methods such as Chain-of-Thought (Wei et al., 2022), Tree-of-Thoughts (Yao et al., 2023), and Self-Consistency (Wang et al., 2023b) have been introduced at inference time (Snell et al., 2025).

However, merely increasing model capacity or reasoning depth does not address the fundamental reliance on knowledge stored within model parameters (Mirzadeh et al., 2024; Huang et al., 2023; Yin et al., 2024). This limitation is evident in tasks requiring external knowledge (Lewis et al., 2020), where LLMs produce uncertainty-laden outputs with degraded quality (Li et al., 2025), in partially observable, decentralized environments where agents' internal memory quickly becomes outdated, amplifying uncertainty and restricting planning performance (Spaan et al., 2006; Amato et al., 2016; 2015).

To the best of our knowledge, no prior work has systematically examined whether uncertainty in embodied planning can be resolved simply by scaling LLMs. Our experiments show that beyond the gains from increasing model capacity or reasoning depth, PCE delivers robust improvements across all tested backbones, indicating that its benefits are additive to and distinct from those of scaling.

**Tree-Structured Reasoning and Search.** Tree-based frameworks have been widely adopted to enhance decision quality, but PCE is conceptually distinct from existing approaches such as Tree of Thoughts (ToT) (Yao et al., 2023) and CoTS (Zu et al., 2025) in both what the tree represents and how communication is treated. ToT constructs a tree over reasoning steps to enhance logical coherence, but it operates primarily within the internal reasoning space of a single agent. It implicitly assumes a fully observable environment and treats tree nodes as cognitive steps rather than probabilistic environmental states, limiting its applicability in dynamic, partially observable multi-agent scenarios. In multi-agent settings, CoTS performs tree search over the joint-reasoning-and-action space and uses iterative communication to explore and prune candidate plans. This communication-centric formulation makes dialogue a prerequisite for search, resulting in increased latency and significant token consumption, especially under imperfect observability.

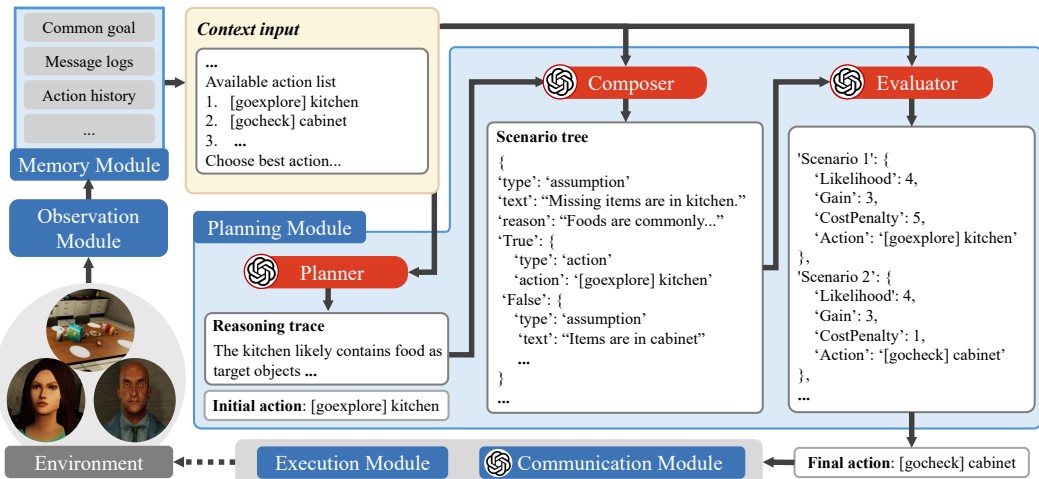

Figure 1: PCE employs a modular architecture with a Planner, Composer, and Evaluator pipeline for planning.

In contrast, PCE structures the decision tree based on uncertain assumptions regarding the environment rather than just reasoning steps. Crucially, PCE treats communication not as the search mechanism itself, but as an atomic action within the search space to be evaluated against physical actions. This allows the agent to select communication only when it yields higher expected utility than acting alone.

## 3 PROBLEM DEFINITION

We model cooperative embodied agents under costly communication as a decentralized partially observable Markov decision process (DEC-POMDP) (Bernstein et al., 2002; Amato et al., 2013). The environment evolves through states $s$ over a finite horizon $t = 1, \ldots, H$. At each time step, $n$ agents execute actions $(a^1, \ldots, a^n)$ with transition probability $P(s' \mid s, a^1, \ldots, a^n)$.

Each agent $i$ receives an observation $o_t^i \in \mathcal{O}^i$, where $\mathcal{O}^i = \mathcal{O}_{\text{env}}^i \cup \mathcal{O}_{\text{com}}^i$ combines direct environmental observations with communication-based observations. Observation histories evolve as $h_t^i = (h_{t-1}^i, o_t^i)$.

The action space $A^i = A_{phy}^i \cup A_{com}^i$ contains both physical and communication actions. A local policy $\pi^i$ maps the current history to an action, $a_t^i = \pi^i(h_t^i)$. Communication actions transmit messages $m_t^i$ constructed from $h_t^i$ and broadcast to all other agents $j \neq i$. We assume costly communication: each $a_t^i \in A_{com^i}$ consumes time that could be used for $A_{phy}^i$, and messages arrive with a one-step delay, i.e. $m_t^i$ appears in $o_{t+1}^i$.

Our objective is to find a set of decentralized policies $\pi = (\pi^1, \ldots, \pi^n)$ that enables the agents to cooperatively achieve a common goal $G$ composed of multiple sub-goals within horizon $H$.

## 4 METHOD

PCE, depicted in Figure 1, adopts the modular design commonly used in embodied agent cooperation (Zhang et al., 2024b; Seo et al., 2025; Liu et al., 2025; Zu et al., 2025). It comprises Observation, Memory, Planning, Communication, and Execution Modules. Among these, we redesign the Planning Module into a Planner–Composer–Evaluator pipeline that explicitly incorporates uncertainty handling, enabling more reliable action selection. Detailed prompting strategies for Planner, Composer, and Evaluator are provided in Appendix A.12.

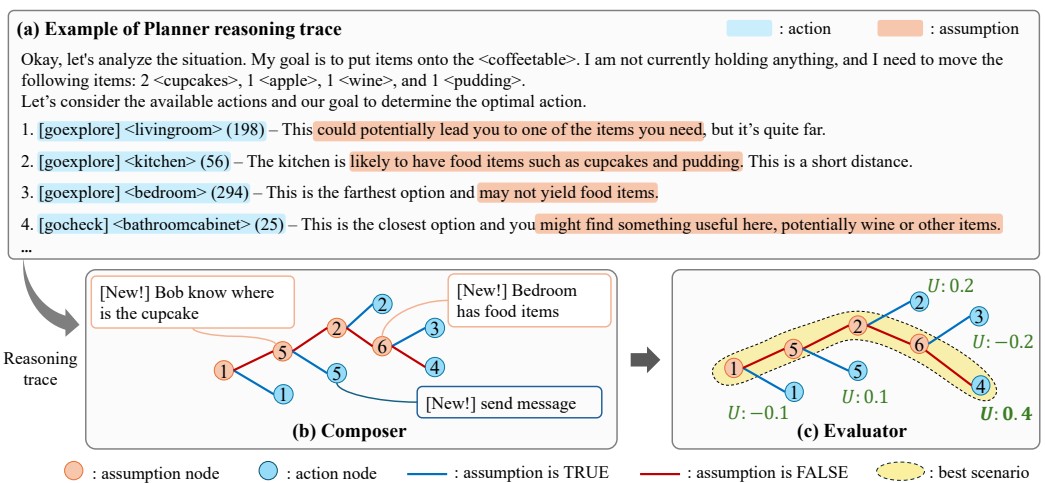

Figure 2: Flow from reasoning trace to action selection. (a) The Planner produces a reasoning trace. (b) The Composer extracts hypotheses from the trace, structures them into a decision tree, and, when needed, generates new assumptions and communication actions to expand unexplored branches. (c) The Evaluator scores each path; The highlighted path indicates the scenario whose leaf node achieves the maximum score ($U$), determining the agent's final selected action.

## 4.1 OBSERVATION AND MEMORY MODULES

The Observation Module transforms raw environmental and collaborator signals into structured perceptual information. Each agent detects object names/IDs, locations, room associations, available interactions, and states of objects within its perceptual range, as well as the collaborator's held objects and position. Closed containers occlude their contents; for example, a cabinet's interior remains unknown until a physical action such as opening is performed, after which newly revealed objects (e.g., a cupcake) become observable. Messages issued by the collaborator in the previous step are also included in the observation.

The Memory Module serves as a unified repository that integrates static information with dynamically updated data. At initialization, it is populated with the common goal $G$ and the low-level action skill book (a set of APIs for performing action). During execution, the memory is incrementally updated with inferred task progress, collaborator information, message logs, and the agent's own previous actions. To prevent unbounded growth in long simulations, message logs are truncated to the most recent $K_{message}$ entries and action histories to the most recent $K_{action}$ entries. See Appendix A.3 for details.

## 4.2 PLANNER: REASONING FOR ACTION SELECTION

The planner forms the first stage of our Planning Module. It receives as *context input* the common goal $G$, current progress, message logs, recent actions from memory, and available action list. Using this, it reasons about why each available action could contribute to achieving $G$ and produces an initial action choice accordingly. For example, if the agent has reached the kitchen and observed a cabinet but not yet interacted with it, the Planner may propose an action such as '[gocheck] <kitchencabinet> (78)'. Leveraging the reasoning capabilities of LLMs, the Planner outputs not only a candidate action but also the associated reasoning trace.

Empirically, each candidate action in this trace tends to be grounded in a single partial assumption about the uncertain environment. For example, as shown in Figure 2-(a), the available action '[gocheck] <bathroomcabinet> (25)' is grounded solely in the assumption that 'you might find something useful here,' while other assumptions relevant to alternative actions remain implicit. These traces therefore reveal isolated assumption-action links but do not show how different assumptions relate to each other, limiting the Planner's ability to rank actions under uncertainty.

### 4.3 COMPOSER: FROM REASONING TRACE TO SCENARIO TREE

The Composer organizes an explicit decision tree. It begins by semantically interpreting the *context input* and the Planner's reasoning trace to explicitly identify key uncertainties (assumptions). Internal nodes represent these assumptions with True/False splits whose branch conditions are inherited along the path. A root-to-leaf path therefore defines a scenario, which is a combination of assumptions culminating in a leaf node. Specifically, each leaf node is assigned an action aimed at handling the uncertainty of the given path—either a physical action for direct interaction or a communication action for sharing information or instructing the collaborator.

Instead of enumerating all assumption-action relationships, the Composer expands the tree top-down. At each node, a local ranking policy selects the next assumption to branch on, prioritizing those that most reduce uncertainty and most strongly influence subsequent action choice. Rather than computing true probabilities, which would amount to solving an intractable POMDP, we approximate these criteria using LLMs' commonsense reasoning. Only assumptions not yet assigned on the path and consistent with current premises are considered, preventing incompatible scenarios from being generated.

When no suitable assumptions remain, the Composer proposes new atomic assumptions grounded in entities already present in the context ($G$, message logs, etc.). These candidates are ranked by the same policy and, if selected, inserted as new branches. Expansion is limited at depth $D$ or stops early when further splits would not materially affect action choice; the resulting node becomes a leaf and is assigned the most appropriate action. Multiple leaves may map to the same action, reflecting its suitability across scenarios, while some initial actions may be dropped if never optimal.

Figure 2-(b) illustrates this process. Given the goal of finding food items, a root assumption 'the living room contains them' leads the True branch toward a '[goexplore] <livingroom> (198)'. On the False branch, the premise changes, and the Composer generates a new assumption that the collaborator (Bob) may know the cupcake's location, shifting the plan from physical exploration to an information gathering action such as [send message].

### 4.4 EVALUATOR: SCENARIO LIKELIHOOD, CONDITIONAL GAIN, AND EXECUTION COST

The Evaluator takes the *context input* and the decision tree. It then scores each leaf to guide action choice under uncertainty. For every root-to-leaf path, it estimates 1) how likely the scenario $\mathcal{S}$ is, 2) how much the action would advance the goal if the scenario holds, and 3) how costly the action is to attempt. All scores are normalized to $[0, 1]$ for comparability.

**Scenario likelihood ($\mathcal{L}$).**  $\mathcal{L}(\mathcal{S})$ is the estimated probability that the premise of the leaf scenario is true, assessed by an LLM against the agent's observation and message history. For instance, if the collaborator was last seen in the kitchen, a scenario stating that the collaborator is still there receives a higher score than one claiming that the kitchen is empty.

**Conditional gain ($\mathcal{G}$).**  $\mathcal{G}(a)$ measures how much executing action $a$ would advance the goal given that the scenario is true. This value is also estimated by an LLM.

Exploiting $\mathcal{L}$ and $\mathcal{G}$, we define the expected gain as

$$\mathbb{E}[\text{gain}] = \mathcal{L}(\mathcal{S}) \cdot \mathcal{G}(a),$$

where $\mathcal{G}(a) = 0$ when the scenario is false.

**Execution cost ($C$).**  $C(a)$ quantifies the immediate resources required to attempt $a$, regardless of its success. We decompose the cost into movement and communication terms using indicator functions:

$$C(a) = \alpha\, d(a)\, \mathbf{1}\{\text{move}(a)\} + \beta\, \ell(a)\, \mathbf{1}\{\text{comm}(a)\},$$

where $d(a)$ is the estimated traversing distance, $\ell(a)$ is the estimated message length, $\alpha, \beta > 0$ are scaling constants that balance the cost of movement and communication, and $\mathbf{1}\{\cdot\}$ is the indicator. This design expresses the mutually exclusive nature of movement and communication: $\mathbf{1}\{\text{move}(a)\} + \mathbf{1}\{\text{comm}(a)\} = 1$.

**Final score ($U$).** For each leaf action, the Evaluator computes a final score:

$$U(\mathcal{S}, a) = \mathbb{E}[\text{gain}] - \lambda\, C(a),$$

where $\lambda > 0$ controlling cost sensitivity. This score integrates scenario probability, goal-directed effectiveness, and execution expense. Ranking leaves by $U(\mathcal{S}, a)$ yields a rational action choice under uncertainty, as shown in Figure 2-(c). The empirical setting of $\alpha, \beta$, and $\lambda$ is discussed in the experiments.

### 4.5 Communication and Execution Modules

These modules translate the Planning Module's decisions into concrete behavior. When a communication action is selected, the Communication Module sends or responds to messages from collaborators. When a physical action (e.g., moving, grasping an object, transporting a target) is selected, the Execution Module carries it out by computing routes with A* search and invoking the appropriate low-level Python APIs from the action skill book.

## 5 Experiment

**Benchmarks.** Under the constraint of partial observability in decentralized control, we conduct experiments on two multi-objective household tasks composed of multiple rooms: C-WAH (Zhang et al., 2024b) and TDW-MAT (Zhang et al., 2024b). C-WAH consists of 10 episodes, where agents are required to accomplish 3–5 sub-goals in each episode, with the horizon $H$ set to 250 steps. TDW-MAT consists of 24 episodes, where agents are tasked with achieving the goal of transporting 10 target objects, with the horizon $H$ set to 3000 steps. Full environment details appear in Appendix A.1.

**Metrics.** In C-WAH, we measure *Total Steps* to evaluate how quickly agents achieve the goal. In TDW-MAT, we measure the proportion of target objects transported by the agents out of the total goal objects, grouped as *Food*, *Stuff*, and their average *Total*.

We also evaluate system efficiency using *Usages* and *Comm*. *Usages* denotes the total token consumption generated by the entire system. This explicitly includes not only communication tokens but also all internal tokens generated by the LLM modules within the framework, serving as a proxy for total computational cost. *Comm* measures the number of communication actions. Unlike other metrics, *Comm* does not have an intrinsic "better is lower" or "better is higher" interpretation. Communication actions are not direct goal completions but context-dependent interventions: in some cases, they slow progress through unnecessary exchanges, while in others, they avert false plans and improve coordination. We therefore treat *Comm* as a descriptive measure reported for diagnostic analysis rather than as a success metric, while primary comparisons focus on task performance and token usage.

**Baselines.** We compared PCE with four representative LLM-based cooperative agent frameworks: CoELA (Zhang et al., 2024b), REVECA (Seo et al., 2025), CaPo (Liu et al., 2025), and CoTS (Zu et al., 2025). CoELA first demonstrated LLM-driven cooperation for embodied agents. REVECA introduced memory management, relative proximity-based planning, and plan validation. CaPo designed plan optimization through iterative debates, and CoTS integrated multi-plan exploration with Monte Carlo Tree Search. All baselines are run under identical environmental and communication settings. For our PCE framework, we use the default hyperparameters $D = 3, \alpha = 1, \beta = 1, \lambda = 1, K_{action} = 10, K_{message} = 3$. Further details of the baselines can be found in Appendix A.2.

**LLMs.** To test robustness across model backbones, each framework is run on three diverse LLMs: *gpt-4o-mini-2024-07-18* (GPT-4o mini) (Hurst et al., 2024), *google/gemma-3-4b-it* (Gemma3:4B) (Team et al., 2025), and *openai/gpt-oss-20b* (GPT-OSS:20B) (Agarwal et al., 2025). GPT-4o mini is a commercial LLM, whereas Gemma3:4B and GPT-OSS:20B are open-source. GPT-OSS:20B is a large reasoning model that performs internal reasoning before generating responses; we use its *medium* reasoning level for comparisons. GPT-4o mini and Gemma3:4B generate outputs without an explicit reasoning module, and we apply zero-shot Chain-of-Thought prompting to encourage higher quality plans.

Table 1: Experimental results in C-WAH. Best result in bold; second-best underlined.

|  |  | PCE | CoELA | REVECA | CaPo | CoTS |
|---|---|---|---|---|---|---|
| GPT-4o mini | *Total Steps* ↓ | **42.76** | 60.40 | 46.80 | 60.82 | 64.00 |
|  | *Comm* | 1.70 | 9.88 | 6.00 | 8.72 | 10.24 |
|  | *Usages* ↓ | 44353.56 | 55467.12 | 46312.16 | **41702.00** | 44628.12 |
| GPT-OSS:20B | *Total Steps* ↓ | **49.60** | 72.72 | 53.86 | 68.34 | 65.26 |
|  | *Comm* | 2.11 | 9.22 | 6.49 | 8.20 | 8.32 |
|  | *Usages* ↓ | **73535.24** | 77727.20 | 74764.24 | 99810.44 | 95428.84 |
| Gemma3:4B | *Total Steps* ↓ | **59.20** | 77.20 | 62.56 | 75.88 | 72.32 |
|  | *Comm* | 3.02 | 9.48 | 9.14 | 7.92 | 4.04 |
|  | *Usages* ↓ | 50984.7 | 49271.24 | **44637.58** | 64015.24 | 51966.64 |

Table 2: Experimental results in TDW-MAT. Best result in bold; second-best underlined.

|  |  | PCE | CoELA | REVECA | CaPo | CoTS |
|---|---|---|---|---|---|---|
| GPT-4o mini | *Total* ↑ | **87.50** | 62.50 | 81.25 | 73.33 | 75.00 |
|  | *Food* ↑ | **89.17** | 65.83 | 80.83 | 82.50 | 84.17 |
|  | *Stuff* ↑ | **85.83** | 59.17 | 81.66 | 64.17 | 65.83 |
|  | *Comm* | 3.58 | 13.33 | 43.76 | 70.79 | 108.92 |
|  | *Usages* ↓ | 197807.29 | **113058.83** | 185453.54 | 281404.71 | 411392.08 |
| GPT-OSS:20B | *Total* ↑ | **81.25** | 55.00 | 73.33 | 65.41 | 59.17 |
|  | *Food* ↑ | **85.00** | 50.83 | 78.33 | 72.50 | 70.83 |
|  | *Stuff* ↑ | **77.50** | 59.17 | 68.33 | 58.33 | 47.50 |
|  | *Comm* | 13.75 | 11.62 | 107.79 | 43.00 | 41.83 |
|  | *Usages* ↓ | 337225.12 | **237498.88** | 370737.17 | 348066.92 | 334912.67 |
| Gemma3:4B | *Total* ↑ | **70.83** | 45.84 | 52.09 | 67.50 | 63.33 |
|  | *Food* ↑ | **71.66** | 50.00 | 56.67 | 70.83 | 64.17 |
|  | *Stuff* ↑ | **70.00** | 41.67 | 47.50 | 64.17 | 62.50 |
|  | *Comm* | 9.42 | 27.42 | 108.00 | 57.88 | 55.96 |
|  | *Usages* ↓ | 184809.08 | **98350.25** | 308221.25 | 217626.50 | 212029.79 |

## 5.1 COMPARATIVE RESULTS

Table 1 and Table 2 summarize the two-agent cooperation results on C-WAH and TDW-MAT. Across all three LLM backbones, our PCE framework consistently achieves the fastest goal completion in C-WAH and the highest success rates in TDW-MAT, outperforming existing approaches on *Total*, *Food*, and *Stuff* metrics.

These gains stem from the agent's ability to act under uncertainty with minimal communication. By explicitly structuring assumptions rather than relying on repeated dialogue, our method spends less time on message exchanges and more time on physical actions. This behavior effectively suppresses unnecessary communication-driven planning cycles, thereby reducing LLM inference cost. Moreover, although PCE's three-module LLM architecture incurs higher per-step inference cost compared with architectures like CoELA that perform two LLM inferences per step, this overhead is offset by PCE's substantial reduction in episode length. Therefore, PCE achieves high performance while maintaining low *Usages* under decentralized partial observability.

In contrast, CoELA, REVECA, CaPo, and CoTS generally rely heavily on communication, which increases the number of steps and delays goal achievement. CoELA lacks systematic communication strategies and mechanisms for evaluating the value of communication, making it difficult to recognize when information exchange is needed to resolve uncertainty in long-horizon, multi-room environments. REVECA can request information required to validate goal-directed plans, but has limited ability to proactively identify highly uncertain factors and query them. CaPo and CoTS have the advantage of generating multi-step plans, but in environments with high uncertainty, these plans often become invalid, leading to repeated replanning.

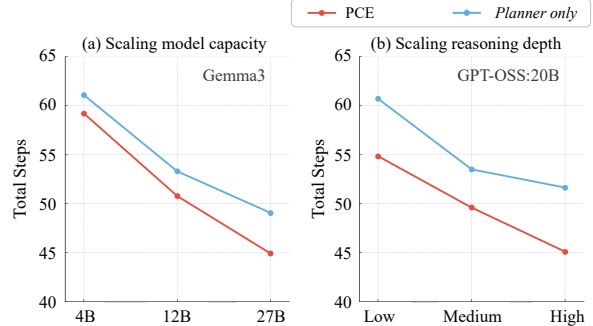

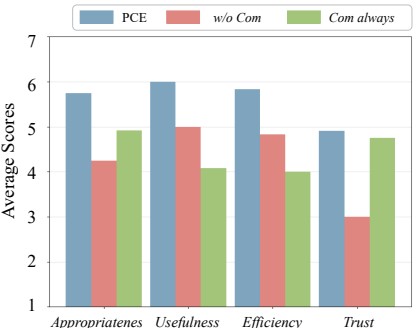

Figure 3: Ablation results about LLM Scaling in C-WAH environment.

Figure 4: User study results in C-WAH environment.

Table 3: Ablation results in C-WAH. Best result in bold; second-best underlined.

|  |  | PCE | *w/o Planner* | *w/o Composer* | *w/o Evaluator* |
|---|---|---|---|---|---|
| GPT-4o mini | *Total Steps* ↓ | **42.76** | 56.46 | 46.82 | 47.34 |
|  | *Comm* | 1.70 | 9.52 | 0.26 | 1.2 |
|  | *Usages* ↓ | 44353.56 | 139918.56 | **33347.66** | 44720.38 |

By contrast, PCE yields stable improvements across all backbones and environments by separating assumption extraction, scenario structuring, and evaluation. This enables the agent to detect when information exchange is genuinely useful, select between physical and communication actions in a principled way, and maintain high performance under decentralized partial observability.

## 5.2 ABLATION STUDY RESULTS

**LLM Scaling.** We first examine whether performance gains come from our structural design rather than simply scaling LLM capacity or reasoning depth. To this end, we compare PCE with a *Planner only* variant that removes the Composer and Evaluator, eliminating explicit uncertainty handling. As shown in Figure 3, even when the backbone size is increased from Gemma3:4B→12B→27B or when GPT-OSS:20B's reasoning depth is raised from Low→Medium→High, *Planner only* shows only modest improvements, while PCE consistently achieves faster goal completion. This indicates that fragmented, implicitly handled assumptions persist under mere scaling, leading to inefficient exploration and occasional planning errors. In contrast, PCE's Composer–Evaluator explicitly organizes and scores these assumptions, demonstrating that our structured uncertainty handling synergizes with scaling to provide a stable performance advantage regardless of the underlying backbone.

**Component Analysis.** We also ablate individual components of the PCE pipeline using GPT-4o mini. As reported in Table 3, removing any module (*w/o Planner*, *w/o Composer*, *w/o Evaluator*) reduces performance. Without the Planner, scenario trees are built directly from context. This increases the difficulty of scenario exploration and often results in incoherent or redundant branches. Without the Composer, the agent relies solely on the Planner's reasoning trace for evaluation, which prevents it from accounting for conflicting or complementary relationships among multiple assumptions and results in incomplete exploration of alternative scenarios. Without the Evaluator, actions are selected without quantitative likelihood–gain–cost assessment, which degrades decision quality. These results confirm that each module contributes essentially to uncertainty-aware planning.

We performed additional analyses to stress-test the structural foundations of PCE and rigorously evaluate its reliability and scalability.

First, we constructed *physical-only* and *communication-only* variants of PCE and conducted comparative experiments, alongside separate comparisons between PCE and reasoning-centric baselines (Chain-of-Thought, Tree-of-Thoughts, Self-Consistency). These analyses confirm that PCE's high

performance does not stem merely from more sophisticated reasoning traces, but from the explicit structuring and evaluation of uncertainty (Appendix A.5).

Second, we assessed the robustness of the framework through a series of quantitative and qualitative evaluations: (1) scalability tests with increasing number of agents to verify cooperative efficiency (Appendix A.9); (2) reliability assessments of the Composer and Evaluator based on human–expert correlation studies (Appendix A.10, A.11); (3) hyperparameter sensitivity analyses to examine configuration stability (Appendix A.5); and (4) comparisons with an MCTS-based planner to demonstrate efficiency advantages over traditional tree search (Appendix A.8).

Finally, we present qualitative case studies illustrating how PCE corrects ill-posed plans through its structured decision-making process, thereby providing an interpretable grounding for the observed quantitative improvements (Appendix A.7).

### 5.3 User Study Results

In real human collaboration, communication is a double-edged sword: excessive messaging disrupts workflow, while the absence of necessary exchanges makes intentions opaque and degrades joint performance. We hypothesized that PCE's ability to structure and evaluate assumptions would allow it to trigger communication only when genuinely useful, producing behavior that humans perceive as aligned, efficient, and trustworthy. To test this, we ran a user study in the C-WAH environment comparing (1) PCE, (2) *w/o Com* (communication actions removed), and (3) *Com always* (communication forced before each physical action). Twelve participants (mean age 26.8; 8 male, 4 female) received the same observations and action choices as the agent.

After completing each method, participants answered four questions on a 7-point Likert scale (1: strongly disagree, 7: strongly agree): 1) "Did the agent perform actions appropriate to your intentions?" (*Appropriateness*), 2) "Was the agent helpful in collaboration?" (*Usefulness*), 3) "Did the agent's performance contribute to achieving the goal efficiently?" (*Efficiency*), and "4) Did you feel a sense of trust with the agent?" (*Trust*). They then joined brief qualitative interviews.

As shown in Figure 4, PCE scored highest across all questions, confirming our hypothesis: by reasoning over uncertainties and initiating communication selectively, PCE achieves a balance of clarity and efficiency that users recognize as more cooperative and reliable. Interview feedback noted that *Com always* disrupted workflows, while *w/o Com* made the agent's intentions unclear and harder to trust. Detailed results are in Appendix A.6.

## 6 Conclusion

This paper presented PCE, a modular Planner-Composer-Evaluator framework that extracts and structures implicit assumptions embedded in LLM reasoning traces, enabling embodied agents to plan under partial observability with minimal communication. Across two multi-objective benchmarks (C-WAH and TDW-MAT) and three diverse LLM backbones, PCE consistently outperformed communication-centric baselines in success rate and task efficiency while showing comparable token usage. Ablation studies confirmed that each module is indispensable and that explicit uncertainty handling unlocks performance gains beyond what model scaling alone provides. A user study further showed that PCE produces communication patterns that humans perceive as efficient and trustworthy.

While our experiments focus on simulated multi-room household tasks, the proposed mechanism is not tied to a specific domain. Future work will explore extending PCE to more complex and dynamic environments, larger and more diverse agent teams, and adaptive discovery of new assumptions in real time. These directions aim to test the framework's scalability and generality under richer uncertainty, advancing the broader goal of principled, decentralized cooperation with embodied agents.

### Acknowledgments

This work was supported by the National Research Foundation of Korea (NRF) grants funded by the Korean government (MSIT) (No. RS-2025-00518643, No. RS-2025-24802983), and by the ICT

Creative Consilience Program through the Institute of Information & Communications Technology Planning & Evaluation (IITP) grant funded by the Korea government (MSIT) (IITP-2026-RS-2020-II201819).

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

## A APPENDIX

### A.1 ENVIRONMENT SETTINGS DETAILS

In our experiments, the agent's observations are obtained through the APIs provided by each environment, following previous studies (Zhang et al., 2024b). We perform experiments in partially observable environments, including C-WAH and TDW-MAT. In the next section, we present the details of these environments.

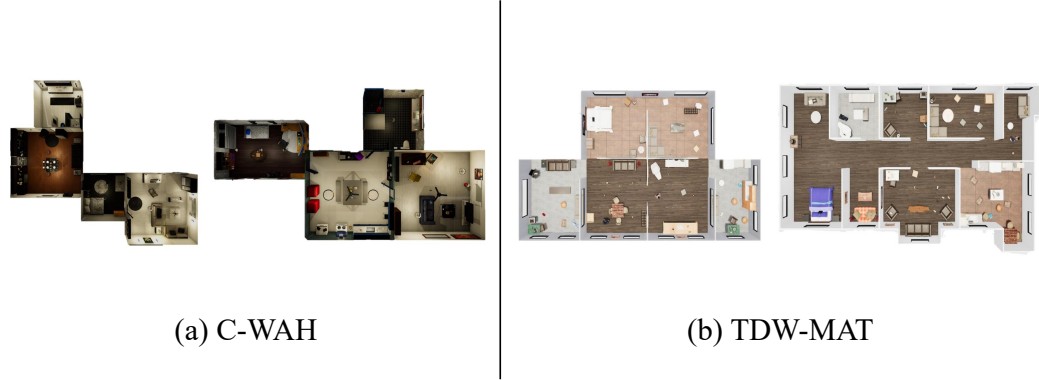

(a) C-WAH           (b) TDW-MAT

Figure 5: example environment

### A.1.1 COMMUNICATIVE WATCH-AND-HELP

The Communicative Watch-And-Help (C-WAH) environment extends the original Watch-And-Help challenge (Puig et al., 2020) by introducing a communication functionality between agents. It is implemented on top of VirtualHome (Puig et al., 2018), a platform for multi-agent simulation. We run 10 episodes in C-WAH, where agents are assigned a common goal to accomplish in each episode. As summarized in Table 4, the common goals fall into five categories, and agents must select from the eight possible actions listed in Table 5 to achieve them.

In this setting, agents can exchange information with each other through communication while executing instructions. When entering a room, an agent can observe all objects that are not inside containers such as fridges or microwaves. To inspect objects within containers, the agent must explicitly perform an additional action to open them. To simulate real-world communication constraints, each agent is restricted to 500 characters per frame.

The horizon $H$ is fixed at 250 simulation steps, and each task contains 3 to 5 subgoals (i.e., objects). Failure to achieve the common goal within 250 steps results in an unsuccessful episode. An illustration of the environment layout is provided in Figure 5-(a).

Table 4: The goal specifications of the C-WAH

| Goals | Description |
|---|---|
| Prepare afternoon tea | put [*cupcake, pudding, apple, juice, wine*] on *coffeetable* |
| Wash dishes | put [*plate, fork*] inside *dishwasher* |
| Prepare a meal | put [*coffeepot, cupcake, pancake, poundcake, pudding, apple, juice, wine*] on *dinnertable* |
| Put groceries | put [*cupcake, pancake, poundcake, pudding, apple, juice, wine*] inside *fridge* |
| Set up a dinner table | put [*plate, fork*] on *dinnertable* |

Table 5: The action space in C-WAH

| Action | Description |
|--------|-------------|
| Walk towards | move to an object in the same room with the agents or a room |
| Turn left | turn left by 30 degrees |
| Turn right | turn right by 30 degrees |
| Grasp | grasp an object |
| Open | open a closed container |
| Close | close an open container |
| Put | put the held objects into an open container or onto a surface |
| Send message | send a message to other agents |

### A.1.2 THREEDWORLD MULTI-AGENT TRANSPORT

The ThreeDWorld Multi-Agent Transport (TDW-MAT) environment, an extended version of the ThreeDWorld Transport Challenge (Gan et al., 2021b), is built on TDW (Gan et al., 2021a). It incorporates more natural object placements and provides a richer set of objects and containers that support item transportation. The common goal in TDW-MAT involves transporting items across two categories: Food and Stuff. Each episode consists of 10 target objects and 2–5 containers, which are strategically placed to enable the transport of multiple items at once, as detailed in Table 6.

Unlike in C-WAH, agents in TDW-MAT cannot acquire complete information about a room without executing a full 360-degree rotation in 15-degree increments. Communication is restricted to 500 characters per frame, and the horizon $H$ is set to 3000 simulation steps. The environment layout is illustrated in Figure 5-(b), and the complete action space is provided in Table 7.

Table 6: The target objects and containers of the TDW-MAT environments.

| Task | Type | Object Name |
|------|------|-------------|
| Food | Containers | *bowl, plate, tea_tray* |
| | Objects | *bread, burger, loaf_bread, apple, banana, orange* |
| Stuff | Containers | *plastic_basket, wicker_basket, wood_basket* |
| | Objects | *iPhone, pen, key, iPod, lighter, purse, calculator, pencil_bucket, mouse* |

Table 7: The action space of the TDW-MAT environment.

| Action | Description |
|--------|-------------|
| Move forward | move forward 0.5m |
| Turn left | turn left by 15 degrees |
| Turn right | turn right by 15 degrees |
| Grasp | grasp an object |
| Put In | put the target into the container |
| Drop | drop the objects held in hand |
| Send message | send a message to other agents |

## A.2 BASELINES

CoELA (Zhang et al., 2024b) implements a modular LLM-driven multi-agent architecture: each agent has perception, memory, planning, communication, and execution modules. Agents exchange linguistic updates and reason over what to do next. It is decentralized and flexible, and with strong LLMs can achieve good multi-agent coordination in embodied tasks.

REVECA (Seo et al., 2025) proposes an LLM-based cooperative agent architecture that leverages information relevance and relative proximity for adaptive planning, and employs trajectory-based validation to prevent false planning caused by collaborators' actions, yielding more efficient memory use and stronger coordination under partial observability.

CaPo (Liu et al., 2025) introduces a collaborative meta-plan generation phase, where agents jointly construct a high-level task decomposition and allocate subtasks before execution. During the task, agents engage in progress-adaptive re-planning, dynamically updating the meta-plan when new information or environmental changes are observed. This two-stage process of structured plan construction followed by adaptive revision enhances coordination, reduces redundancy, and improves task efficiency in embodied multi-agent settings.

CoTS (Zu et al., 2025) introduces a collaborative tree search framework for multi-agent planning, inspired by Monte Carlo Tree Search. Instead of following a single plan path, agents explore multiple candidate branches, evaluate them using LLM-based reasoning or heuristic scoring, and then converge on a coherent long-term strategy. A dedicated plan evaluation module monitors execution progress and selectively triggers plan updates when necessary, striking a balance between adaptability and stability. This design provides greater foresight, improves resilience to planning errors, and reduces unnecessary disruptions during collaboration.

## A.3 EXAMPLE OF MEMORIES IN MEMORY MODULES

In this session, we describe the information stored in the memory module—including the common goal, object information, collaborator information, message logs, previous actions, and the low-level action skill book—and provide detailed explanations along with illustrative examples for each.

**Common goal.** The common goal $G$ is expressed in natural language as a description of the household task that the agents must jointly complete in the given episode. Below are examples of common goals in C-WAH and TDW-MAT.

- **C-WAH**: "Find and put target objects 1 pudding, 1 juice, 1 apple, 2 cupcakes onto the goal location <coffeetable> (268)."
- **TDW-MAT**: "Transport 2 breads, 2 burgers, 4 apples, 1 loaf of bread, 1 banana to the bed."

**Object information.** Object IDs and names, 3D positions, room IDs and names, available actions, and object state (e.g., whether the object is held, inside a container, or available to be grasped). An example of this is shown in Listing 1.

**Collaborator information.** Collaborator's held objects and 3D position. This information is updated when the collaborator enters the observation range, and the module stores the most recently observed values.

**Message logs.** The messages sent by all the agents. Only the most recent $K_{message}$ messages are stored, and in our implementation we set $K_{message} = 3$.

**Previous Actions.** The actions performed by the agent from the past to the present are stored. Only the most recent $K_{action}$ actions are kept, and in our implementation, we set $K_{action} = 10$.

**Low-level action skill book.** The low-level action skill book outlines the Python APIs required for interacting with the environment. An example of this is shown in Listing 2.

Listing 1: Example of object information

```
1  # A simple example of an object information list with one entry.
2  object_information_list = [{
3      "object_id": 21,
4      "object_name": "apple",
5      "position": [13.22, 1.20, 5.41],
6      "available_action": "gograb",
7      "room_name": "livingroom",
8      "room_id": 198,
9      "states": [15, "GRABBABLE"]
10 }]
```

Listing 2: Example of low-level action skill book

```
1  # A simple example of a low-level action skill book
2  def goexplore(self):
3      if current_room == target_room:
4      # Move towards a specific room based on the plan.
5
6  def gocheck(self):
7      if 'OPEN' in container['states']:
8      # Check the status of a container and attempt to interact with it.
9
10 def gograb(self):
11     if target in reachable_objects:
12     # Attempt to grab an object, ensuring availability and conditions.
13
14 def goput(self):
15     if len(grabbed_objects) > 0:
16     # Place the grabbed object in the specified location or container.
```

A.4    DISTANCE-BASED PLANNING

When only an *available action list* is provided, LLM agents often struggle to account for efficient movement paths or division of labor that enhances cooperative efficiency during planning. Therefore, following prior studies, we also employ physical distance as a key cognitive factor. Specifically, the Memory Module computes the agent–object distance and collaborator–object distance based on the location of the target object for each available action, the most recently observed collaborator position, and the agent's own position. These distance annotations are incorporated into the textual descriptions of available actions, thereby guiding the Planner to consider both spatial efficiency and task allocation when selecting an action. Ultimately, the Planner takes the augmented text prompt as input and selects the action from the *available action list* that is most suitable for achieving the common goal.

A.5    ADDITIONAL ABLATION STUDY RESULTS

**Uncertainty Handling.**    We analyze the cost–gain trade-off between physical actions and communication actions, which employ contrasting strategies for handling uncertainty. To this end, we compare four baselines: PCE, *Phy-act only*, *Com-act only*, and *Planner only*. Specifically, *Phy-act only* relies solely on physical actions by the Composer to resolve uncertainty, while *Com-act only* employs only communication actions. For all baselines, we use GPT-4o mini as the backbone LLM. The results, presented in Table 8-(Uncertainty Handling), demonstrate that PCE achieves the joint goal with fewer *Steps* than the other baselines. In particular, *Phy-act only* eliminates communication costs entirely, but must compensate through physical exploration, resulting in excessive movements and delayed goal achievement. In the *Com-act only* condition, the agent tries to prioritize resolving all uncertainties through communication before executing any physical actions. Specifically, the Evaluator first inspects all scenarios in the decision tree that can be clarified via

Table 8: Ablation results in C-WAH. Best result in bold; second-best underlined.

| | | PCE | *Com-act only* | *Phy-act only* | *Planner only* |
|---|---|---|---|---|---|
| Uncertainty Handling | *Total Steps* ↓ | **42.76** | 46.69 | 45.38 | 46.69 |
| | *Comm* | 1.70 | 4.78 | 0.0 | 2.39 |
| | *Usages* ↓ | 44353.56 | 49051.36 | 45536.14 | **26678.88** |
| | | PCE | *CoT* | *ToT* | *SC* |
| Reasoning Enhancements | *Total Steps* ↓ | **42.76** | 55.40 | 50.96 | 54.86 |
| | *Comm* | 1.70 | 6.56 | 1.22 | 1.94 |
| | *Usages* ↓ | **44353.56** | 130633.3 | 149666.80 | 129134.10 |

Table 9: Hyperparameter sensitivity analysis results.

| | Variant 1 | Total Steps | Variant 2 | Total Steps |
|---|---|---|---|---|
| **Tree Max Depth** ($D$) | $D = 2$ | 44.6 | $D = 4$ | 42.4 |
| **Cost Weight** ($\alpha$) | $\alpha = 0.5$ | 50.1 | $\alpha = 1.5$ | 45.5 |
| **Cost Weight** ($\beta$) | $\beta = 0.5$ | 48.6 | $\beta = 1.5$ | 44.6 |
| **Global Penalty** ($\lambda$) | $\lambda = 0.5$ | 58.7 | $\lambda = 1.5$ | 45.4 |
| **Memory History** ($K_{action}$) | $K_{action} = 5$ | 44.4 | $K_{action} = 15$ | 44.3 |
| **Memory History** ($K_{message}$) | $K_{message} = 2$ | 49.5 | $K_{message} = 4$ | 49.7 |

communication. After each communication action, the PCE framework is invoked again, and the Composer reconstructs the decision tree to examine whether additional information can be obtained through dialogue. This iterative process continues until the Composer determines that no further meaningful information can be acquired from the collaborator. Only when no scenario in the decision tree can yield additional benefits through communication does the agent proceed to perform physical actions. Consequently, *Com-act only* enables rapid acquisition of collaborator information, but redundant communication delays goal completion and increases token costs. Moreover, *Planner only*, lacking any explicit uncertainty handling, exhibits the lowest overall cooperative performance. Taken together, these findings indicate that approaches biased toward a single mode of uncertainty resolution suffer from inherent structural limitations in cost–efficiency. By contrast, PCE leverages both strategies in a balanced manner through the Composer–Evaluator, thereby minimizing costs while simultaneously enhancing uncertainty handling and cooperative performance.

**Reasoning Enhancements.** We analyze the effectiveness of different reasoning enhancement strategies in cooperative planning under partial observability. To this end, we compare four baselines: PCE, Chain-of-Thought (*CoT*), Tree of Thought (*ToT*), and Self-Consistency (*SC*). Specifically, *CoT* performs planning through a single linear chain of thought, *ToT* expands reasoning into multiple branches using a tree of thought and explores candidates via tree search, and *SC* applies self-consistency by sampling multiple reasoning traces and aggregating them through majority voting. For all baselines, we use GPT-4o mini as the backbone LLM. The results, presented in Table 8-(Reasoning Enhancements), demonstrate that PCE consistently achieves the joint goal more efficiently and reliably than the reasoning-only baselines. In particular, *CoT* provides stable reasoning but lacks diversity, often leading to brittle plans under uncertainty. *ToT* increases diversity by exploring multiple branches, but incurs significant token costs and sometimes selects suboptimal branches due to the absence of explicit uncertainty evaluation. *SC* improves robustness by aggregating multiple sampled plans, yet fails to distinguish between conflicting assumptions, resulting in inconsistent decision-making. Taken together, these findings indicate that simply enhancing reasoning through scale or sampling is insufficient for cooperative planning under uncertainty. Moreover, the absence of explicit structuring and evaluation of uncertainty can cause erroneous assumptions to be amplified as reasoning depth increases, thereby leading to the accumulation of hallucinations and the propagation of errors. By contrast, PCE explicitly structures and evaluates assumptions through the Composer–Evaluator, enabling principled uncertainty-aware planning that balances efficiency, robustness, and cost-effectiveness.

Table 10: Representative interview responses from user study participants.

| | **Participant Feedback** |
|---|---|
| *w/o Com* | "There was no conversation at all, which was convenient in some sense, but it did not feel like collaboration."
"Since the agent did not speak, it was difficult to know what it was doing, and I could not trust it."
"There was no collaboration at all." |
| *Com always* | "The agent talked too much, and my workflow was constantly interrupted."
"Frequent communication had some benefits, but constant talking was uncomfortable."
"Division of labor was easier, but frequent conversations made it hard to concentrate and felt frustrating." |
| **PCE** | "There was no unnecessary communication, and I did not feel any inconvenience."
"The amount of communication was appropriate, and the agent answered kindly when asked."
"I think the agent worked efficiently and effectively." |

**Hyperparameter Sensitivity Analysis.** To examine how each hyperparameter influences planning depth, communication frequency, and execution efficiency, we conducted a comprehensive sensitivity analysis on the C-WAH benchmark. Each parameter was varied independently while keeping all others fixed. The full numerical results are provided in Table 9. Under the default configuration ($D = 3, \alpha = 1, \beta = 1, \lambda = 1, K_{action} = 10$, and $K_{message} = 3$), the model achieves 42.76 total steps. A shallower tree ($D = 2$) limits exploration of alternative assumption scenarios, resulting in degraded performance. Increasing the depth ($D = 4$) provides only marginal gains (42.4 steps) while incurring exponential tree growth in the worst case that increases hallucination risk and computational cost. These results identify $D = 3$ as an effective balance between expressiveness and robustness. The coefficients $\alpha$ and $\beta$ control the relative impact of physical and communicative action costs. Lower $\alpha$ or higher $\beta$ causes the agent to underestimate physical actions, leading to excessive exploration. Conversely, higher $\alpha$ or lower $\beta$ biases decisions toward communication-heavy strategies, also degrading performance. The default values ($\alpha=1, \beta=1$) provide a balanced weighting across modalities. The global penalty $\lambda$ regulates the trade-off between expected utility and execution cost. A smaller value overemphasizes likelihood and gain, often pushing the agent toward optimistic yet risky branches. A larger value enforces overly conservative behavior. The empirical trend shows that $\lambda=1$ provides a stable middle ground. Varying the memory-history windows ($K_{action}, K_{message}$) shows that both shorter and longer lengths negatively impact performance. Shorter histories weaken long-horizon consistency, while longer histories incorporate irrelevant context and inflate input size. Sensitivity is particularly pronounced for $K_{message}$, where deviations from the default disrupt coordination under partial observability.

A.6 ADDITIONAL USER STUDY RESULTS

To conduct comparative analyses, we first tested the normality of the data using Shapiro–Wilk and Kolmogorov–Smirnov tests at a 5% significance level. As the results showed that the data were not normally distributed, we employed the Wilcoxon signed-rank test for pairwise comparisons. The results are presented in Figure 6.

The analysis revealed that PCE significantly outperformed *w/o Com* in terms of *Trust* and *Appropriateness*, indicating that appropriately timed communication improved user trust and alignment with user intentions. Furthermore, PCE achieved significantly higher scores than *Com always* in *Usefulness* and *Efficiency*, suggesting that excessive communication hindered collaborative efficiency, while our approach enhanced cooperation by providing information only when necessary.

In addition, to provide a richer account, we report selected interview responses from the user study in Table 10. Participants consistently noted that under the *w/o Com* condition, the absence of di-

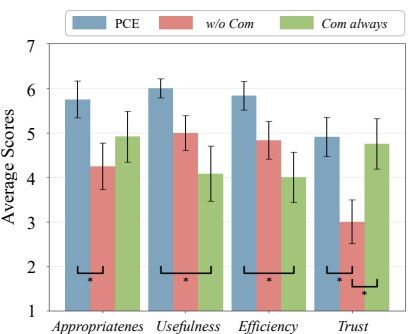

| | | Total Steps ↓ |
|---|---|---|
| | PCE | **72.42** |
| GPT-4o mini | *w/o Com* | 75.67 |
| | *Com always* | 114.25 |

Table 11: User study results in C-WAH environment. Best result in bold; second-best underlined.

Figure 6: User study results with C-WAH. This figure illustrates the mean scores and associated standard errors for responses to four research questions. Statistical significance was denoted as * for $p < 0.05$.

alogue made it difficult to feel as though they were collaborating with the agent, which in turn reduced their trust. Under the *Com always* condition, participants reported that frequent questions and information requests disrupted the workflow and delayed goal completion. In contrast, with PCE, participants stated that they were able to receive necessary information through appropriately timed dialogue, which facilitated faster goal achievement and resulted in minimal discomfort despite interacting with an agent.

As shown in Table 11, PCE required fewer *Total steps* than both baselines. In the *w/o Com* condition, the absence of dialogue did not consume time for communication, but the lack of information sharing led to substantially more steps to achieve the goal, resulting in lower efficiency. In contrast, the *Com always* condition increased the number of steps due to frequent interruptions in the workflow caused by excessive communication. By comparison, PCE leveraged timely communication to obtain necessary information and reduce redundant movements, thereby enabling more efficient task completion.

The results of the user study revealed that communication actions go beyond mere information exchange, serving as a key factor in shaping how human users experience and trust their interactions with agents. While unnecessary communication hindered collaborative efficiency, contextually appropriate communication accelerated goal achievement and strengthened trust in the collaboration.

### A.7 QUALITATIVE CASE STUDIES

In this section, we present qualitative case studies to illustrate how the PCE framework operates in practice. Specifically, we analyze scenarios where the baseline Planner suggests suboptimal or illogical actions due to hallucinations, and demonstrate how PCE corrects these through structured assumption generation and multi-criteria evaluation. These cases highlight PCE's versatility in handling spatial uncertainty, social inference, and cost-benefit analysis.

**Case 1: Correction of Illogical Physical Search** **Scenario:** Agent Alice has successfully collected most items, with only one target item, a *loaf_bread*, remaining. She is currently in the *Livingroom* (ID: 4000). The *Bedroom* (ID: 8000) has already been thoroughly searched and found to contain only a bed (no target items). Meanwhile, the *Kitchen* (ID: 5000) and *Office* (ID: 7000) remain completely unexplored.

**Planner Failure:** Relying on a simple heuristic that the remaining item must be somewhere, the Planner hallucinates a potential gain in the previously visited room and proposes returning to the *Bedroom* to search again, ignoring the negative observation history.

**PCE Resolution:** The Composer identifies the core uncertainty: "Where is the remaining loaf_bread located?" It constructs a scenario tree with competing assumptions: (A) "The bread is in the Bed-

room" and (B) "The bread is in the Kitchen." The Evaluator then scores these paths. The path containing Assumption A receives an extremely low *Scenario likelihood* score because the history log confirms the Bedroom was already checked. Conversely, Assumption B is assigned a high *Scenario likelihood* (as the area is unexplored) and high *Conditionl gain*. Consequently, PCE overrides the Planner's decision and selects `[goexplore] <Kitchen>`, ensuring logical consistency in the search strategy.

**Case 2: Optimizing Transport Efficiency**    **Scenario:** Alice is in the *Livingroom* holding a target item. The final drop-off location (Coffee Table) is within her immediate vicinity. The *Kitchen*, which typically contains many objects, is far away.

**Planner Failure:** Driven by a commonsense prior that "Kitchens contain food," the Planner suggests navigating to the distant *Kitchen* to gather more items before returning. This is greedy and inefficient because (1) Alice can only carry limited items, and (2) the travel cost is high.

**PCE Resolution:** The Composer expands the reasoning trace by introducing a critical assumption not present in the Planner's trace: "Collaborator Bob may have already cleared the Kitchen." The Evaluator weighs the scenarios. The path to the Kitchen incurs a high *Execution cost* (travel time) and carries a risk that the *Conditionl gain* is zero (if Bob took the items). In contrast, the path leading to the immediate drop-off has zero risk and guaranteed *Conditional gain* for the current item. PCE thus pivots the action to `[goput]`, prioritizing the globally optimal behavior of securing the current point over uncertain exploration.

**Case 3: Strategic Resource Management**    **Scenario:** Agent Bob has finished searching the *Bedroom* (ID: 5000) and is currently in the *Livingroom* (ID: 4000). He has spotted two containers (e.g., a *wood_basket*) but no target items yet. Other rooms remain unexplored. The containers are located far away (210 steps), while an unexplored section of the *Livingroom* (ID: 3000) is much closer (140 steps).

**Planner Failure:** The Planner focuses on the long-term utility of carrying more items and decides to execute `[gograsp] <wood_basket>`. However, it fails to account for the immediate travel cost or the uncertainty of whether the container will actually be useful given the unknown item locations.

**PCE Resolution:** The Composer explicitly questions the utility of the tool, branching into scenarios where "The container aids collection" versus "Direct exploration yields items faster." The Evaluator conducts a cost-benefit analysis. It determines that investing 210 steps for a container of uncertain utility is less efficient than investing 140 steps to resolve the high entropy of the nearby unexplored area. As a result, PCE selects `[goexplore] <Livingroom> (3000)`, favoring immediate information gain and lower cost over a speculative long-term investment.

Table 12: Comparison of Average Total Steps on the C-WAH Benchmark. Best result in bold.

| | Total Steps ($\downarrow$) |
|---|---|
| MHP (MCTS-based) | 64.90 |
| **PCE** | **42.76** |

## A.8 ADDITIONAL BASELINES: COMPARISON WITH MCTS-BASED PLANNER

To provide a comprehensive evaluation of planning efficiency, we compare PCE against a traditional tree-search baseline: the MCTS-based Hierarchical Planner (MHP). Adapted from the approaches in the Watch-And-Help challenge (Puig et al., 2020), MHP utilizes MCTS to find optimal action trajectories. It functions by minimizing graph traversal costs and step counts, optimizing plans primarily based on the topological connectivity of the environment and predefined action costs.

**Analysis.**    As presented in Table 12, PCE demonstrates superior efficiency, completing tasks with significantly fewer steps compared to MHP. While MHP is effective at minimizing traversal costs based on the topological structure, it struggles with the high uncertainty inherent in partially observable settings. Relying on topological connectivity and action costs, MHP tends to perform inefficient exhaustive searches when target locations are semantically ambiguous. In contrast, PCE

leverages the semantic reasoning capabilities of LLMs to explicitly model likelihoods from context (e.g., inferring that an *apple* is likely in the *kitchen* without physical verification) and prioritizes actions based on expected utility rather than proximity alone. These results highlight that in embodied DEC-POMDPs, resolving semantic uncertainty through reasoning is a more critical bottleneck for efficiency than topological path optimization.

## A.9 SCALABILITY ANALYSIS: IMPACT OF NUMBER OF AGENTS

A fundamental challenge in multi-agent planning is the potential for combinatorial explosion as the number of agents ($N$) increases. In this section, we analyze the scalability of PCE from both structural and empirical perspectives, demonstrating how the framework manages complexity in teams larger than two agents.

**Structural Efficiency via Semantic Aggregation.** Extracting uncertainty directly from the full joint state of all agents would lead to combinatorial complexity in decision tree construction. In contrast, our Composer is designed to extract uncertainty solely from the Planner's reasoning trace, observations, and memory. Because the Planner inherently filters out irrelevant details to focus on salient uncertainties, the Composer's search space scales with the complexity of the uncertainty itself, rather than linearly with the number of agents.

We observe that LLMs naturally employ a semantic aggregation when reasoning about multiple collaborators. Rather than generating separate assumptions for each individual agent (e.g., "Did Agent A check?" vs. "Did Agent B check?"), the Planner tends to abstract these into collective predicates (e.g., "Has *any* collaborator checked the kitchen?"). Because the decision tree is constructed based on these aggregated semantic assumptions, a max tree depth (e.g., $D = 3$) remains sufficient to model critical uncertainties without becoming intractable, even as the number of agents grows.

**Empirical Validation** ($N = 3, 4$)**.** To empirically validate this scalability, we extended our experiments on the C-WAH benchmark to include scenarios with 3 and 4 agents, maintaining the same tree depth hyperparameter ($D = 3$).

Table 13: Performance of PCE with increasing number of agents on C-WAH. Total Steps decrease as agents are added, indicating effective coordination without planning overhead.

| Number of Agents ($N$) | Total Steps ($\downarrow$) |
|---|---|
| $N = 2$ | 42.76 |
| $N = 3$ | 34.60 |
| $N = 4$ | 28.50 |

As shown in Table 13, the average Total Steps required to complete the task decreased monotonically as the number of agents increased from 2 to 4. This improvement indicates that PCE successfully coordinates labor division among a larger team without being overwhelmed by planning complexity. The framework effectively converts the increased manpower into task efficiency.

**Limitations.** While PCE scales robustly to small-to-medium-sized teams (up to $N = 4$), we acknowledge that simple semantic aggregation may face limitations in scenarios involving massive numbers of agents or tasks with strict sequential dependencies between heterogeneous agents. We leave the exploration of such extreme scaling conditions to future work.

## A.10 ACCURACY AND STABILITY OF LLM-GENERATED SCORES

A critical component of PCE is the Evaluator's ability to assign reliable scores to generated scenarios. To quantitatively verify the stability of these LLM-generated scores and address concerns regarding the reliability of LLM-as-a-Judge, we conducted a quantitative evaluation.

**Experimental Setup.** We sampled execution logs from 10 tasks in the C-WAH benchmark. Four domain experts, familiar with the benchmark mechanics, served as annotators. These experts were

provided with the same context inputs and scenario trees as the Evaluator and were asked to manually assign scores on a discrete scale of 1 to 5 for three criteria: *Scenario likelihood*, *Conditional gain*, and *Execution cost*.

**Quantitative Results.** We measured the agreement between human annotations and the Evaluator's outputs using Mean Absolute Error (MAE).

Table 14: Mean Absolute Error (MAE) between Human Experts and the LLM Evaluator on a 5-point scale. Low MAE values indicate strong alignment between the model's judgment and expert intuition.

|  | **MAE** |
| --- | --- |
| *Scenario likelihood* | 0.91 |
| *Conditional gain* | 1.10 |
| *Execution cost* | 0.88 |

As shown in Table 14, the Evaluator demonstrates strong alignment with human judgment. The *Execution cost* metric shows the highest precision (0.88), likely because physical distance and time are objective measures. *Scenario likelihood* also shows high agreement (0.91), as it relies on task progress stored in the agent's memory to judge the rationality of each scenario. While *Conditional gain* exhibits a slightly higher error (1.10), this is within an acceptable margin given the inherently subjective nature of estimating future utility in partial observable settings.

**Correlation with Model Scaling.** Furthermore, as illustrated in our main experiment results (see Table 1, Table 2), stronger backbone models (e.g., GPT-4o-mini) consistently yield superior planning outcomes compared to smaller models (e.g., Gemma). This trend confirms that as the reasoning capability of the underlying LLM improves, the precision of the Evaluator's scoring increases, directly translating to higher system stability and more rational decision-making.

## A.11 RELIABILITY ANALYSIS OF THE COMPOSER MODULE

The Composer takes the Planner's free-form reasoning trace as input, restructures it into explicit assumptions, and uses them to construct a decision tree. Therefore, its reliability is critical: if the Composer extracts incorrect information or generates hallucinations, the subsequent decision tree becomes invalid. To rigorously assess this reliability, we conducted two complementary analyses: 1) a quantitative stress test at the assumption level, and 2) an expert evaluation at the decision-tree level. The first experiment focuses on how effectively the Composer identifies critical assumptions even when the Planner's reasoning trace is vague. The second experiment evaluates whether these assumptions are organized into a coherent decision-tree structure.

### A.11.1 QUANTITATIVE STRESS TEST AT THE ASSUMPTION LEVEL

To evaluate whether the Composer reliably identifies critical assumptions, including in cases where the Planner's reasoning trace is vague or incomplete, we designed a quantitative stress test at the level of individual assumption nodes. Since the correctness of extracted or generated assumptions is inherently semantic and context-dependent, it cannot be reliably validated by rule-based metrics or environment rewards alone. Human experts are therefore required to serve as the only trustworthy reference for judging whether an assumption genuinely reflects the Planner's intent or constitutes a reasonable hypothesis in ambiguous contexts. The primary outcome metric for this experiment is the proportion of assumptions judged valid by human experts, which we refer to as the Composer's hit rate.

**Experimental Setup** The experiment utilized execution logs collected from 10 C-WAH tasks. For each log, we modified the Composer's prompt so that every assumption node explicitly self-labels as one of the following types:

- **Extracted**: Assumptions directly grounded in the Planner's reasoning trace.

- **Generated**: Assumptions newly created based on the environment, goal, and memory information.

Next, four domain experts categorized each instance into easy, medium, or hard based on trace clarity. **Easy** cases contain assumptions explicitly stated with minimal noise. **Medium** cases involve assumptions that are inferable but embedded in ambiguous or cluttered reasoning. **Hard** cases correspond to vague traces where the critical assumption is not directly mentioned, requiring implicit inference or hypothesis generation.

Using this categorized dataset, each assumption was independently evaluated by three additional experts. **Extracted assumptions** were considered valid only if explicitly supported by the trace, while **Generated assumptions** were considered valid if they formed a contextually plausible hypothesis given the environment, goal, and agent memory, and collaborator state. Final labels were determined by majority vote among the three evaluators. The resulting hit rate, reported separately for each difficulty level, quantifies how consistently the Composer's assumptions align with human judgment.

Table 15: Composer assumption validity across difficulty levels.

|  | **Easy** | **Medium** | **Hard** | **Overall** |
|---|---|---|---|---|
| Hit Rate (%) | 84.3 | 77.8 | 76.7 | 80.6 |

**Composer Hit Rate Results.** Table 15 summarizes the Composer's assumption validity across different difficulty levels. Overall, the Composer achieved a validity rate of 80.6%, and it maintained a hit rate of 76.7% even in the Hard cases. Although explicit cues diminish and implicit inference becomes increasingly necessary from Easy to Hard, the performance decreases only gradually rather than collapsing. This pattern shows that the Composer is not merely producing plausible-sounding text but is instead reliably identifying uncertainties that are genuinely relevant to the task. It also demonstrates that even when the Planner's reasoning trace contains noise, omissions, or ambiguity, the Composer can still generate assumptions that remain contextually appropriate and aligned with the agent's goals. In summary, the results of this stress test demonstrate that the Composer remains robust even when the quality of its input reasoning trace is degraded.

**Additional Evaluation for System-level Stability.** Additionally, because assumption-level accuracy alone does not fully explain system-level stability, we conducted a complementary analysis using the same dataset to examine how effectively the Evaluator suppresses erroneous assumptions. We separated scenarios into two categories: those in which all assumptions were judged valid by experts, and those that contained at least one invalid assumption. For each scenario, we compared the Evaluator's assigned *Scenario likelihood* and *Final score*.

Table 16: Evaluator scores for scenarios with and without invalid assumptions.

|  | **Scenario likelihood** | **Final score** |
|---|---|---|
| Valid assumptions only | 3.28 | 3.63 |
| Contains invalid assumptions | 2.85 | 2.51 |

Table 16 reports the average *Scenario likelihood* and *Final score* produced by the Evaluator for each group. Scenarios containing invalid assumptions show clear degradation in both metrics. The *Scenario likelihood* drops from 3.28 to 2.85, indicating that the Evaluator detects inconsistencies or contradictions in the underlying knowledge state. The *Final score* shows an even larger decline, from 3.63 to 2.51, which reflects the Evaluator's active suppression of these flawed scenarios during action selection. These results demonstrate that PCE is not a fragile system that relies on perfect assumption extraction from the Composer. Instead, the Evaluator functions as a structural safety layer that prevents unintended errors from propagating into the final action decision, thereby maintaining stable overall performance.

Table 17: Expert evaluation of composer-generated decision trees (7-point Likert Scale).

| Metric | Composer | Human | Evaluation Focus |
|---|---|---|---|
| Q1: Extraction Accuracy | 6.27 | 6.64 | Does the extracted assumption actually exist in the trace? |
| Q2: Generation Capability | 6.39 | 6.73 | Are new assumptions generated appropriately when the trace is vague? |
| Q3: Ranking Logic | 5.83 | 6.23 | Are nodes ordered by uncertainty reduction & impact? |
| Q4: Logical Consistency | 6.18 | 6.55 | Are there contradictions in the decision-tree? |
| Q5: Action Appropriateness | 5.98 | 6.15 | Do leaf actions match the scenario path? |

### A.11.2 EXPERT EVALUATION AT THE DECISION-TREE LEVEL

While the assumption-level analysis shows how accurately the Composer constructs individual nodes, it does not by itself guarantee that these nodes form a coherent strategy when assembled into a full decision tree. This limitation is particularly important in multi-agent planning: even when each assumption is locally correct, the agent may still choose suboptimal actions if the assumptions are misprioritized, placed at inconsistent levels of abstraction, or misaligned with corresponding actions. To assess this risk, we evaluated the structural and logical quality of the decision trees produced by the Composer by comparing them against those constructed by human experts.

Using the same execution logs from the 10 C-WAH tasks employed in the stress test, four domain experts independently constructed decision trees based on identical inputs (environment state, goal, and the Planner's reasoning trace). These human-generated trees were then combined to form a Decision Trees Dataset, and we conducted a blind cross-review on this dataset using a 7-point Likert scale across five evaluation dimensions.

The results are presented in Table 17. While it is expected that human experts achieve slightly higher scores overall, the Composer attains scores of 5.83 or higher across all dimensions, indicating a level of quality that is broadly comparable to expert performance. The strong results in Generation Capability and Logical Consistency show that the Composer not only faithfully reflects the Planner's reasoning but also generates new assumptions when necessary while maintaining coherence with previously established assumptions. The Ranking Logic scores illustrate that the Composer effectively prioritizes key uncertainties, enabling efficient uncertainty reduction during decision making. The Action Appropriateness results further suggest that the Composer correctly interprets the semantics of each scenario path and assigns suitable actions to the corresponding leaf nodes.

## A.12 PROMPT TEMPLATE

---

**Communication**

I'm $AGENT_NAME$. My collaborator $OPPO_NAME$ just sent a message and I must reply concisely and helpfully to keep us coordinated on the shared goal.
Goal: $GOAL$
My Progress: $PROGRESS$
Dialogue history:
$DIALOGUE_HISTORY$
Previous actions: $ACTION_HISTORY$
Last incoming message from $OPPO_NAME$:
$LAST_MESSAGE$

First, directly ANSWER any explicit question in the last message.

Then, provide exactly ONE short follow-up (status, location, or confirmation) that reduces uncertainty most relevant to the goal (e.g., where an item is, who will grab it, whether a room was checked).

If making/confirming commitments, reference specific objects with their full notation <name> (id) ONLY if that exact id is known from context; otherwise use just the name without inventing ids.

Keep it brief. Be polite, action-oriented, and unambiguous.

OUTPUT FORMAT: return exactly one action string in this format:
[send_message] <'...'>
Do not add any extra text before or after.

Answer:

---

Figure 7: Prompts template for Communication.

---

**Planner**

I'm $AGENT_NAME$. I'm in a hurry to finish the housework with my friend $OPPO_NAME$ together. Given our shared goal, dialogue history, and my progress and previous actions, please help me choose the best available action to achieve the goal as soon as possible. Do not choose actions related to goals that have already been achieved or are likely to have been achieved by the opponent. Note that I can hold two objects at a time. All objects are denoted as <name> (id), such as <table> (712).
Goal: $GOAL$
Progress: $PROGRESS$
Dialogue history:
$DIALOGUE_HISTORY$
Previous actions: $ACTION_HISTORY$
Available actions:
$AVAILABLE_ACTIONS$
Answer:

---

Figure 8: Prompts template for Planner.

## Composer

I'm $AGENT_NAME$. I'm examining my reasoning trace for uncertainties before acting.
Reasoning Trace = the planner's internal chain-of-thought used to select the next action in the previous planning step.
Goal: $GOAL$
Progress: $PROGRESS$
Dialogue history:
$DIALOGUE_HISTORY$
Previous actions: $ACTION_HISTORY$
Available Actions:
$AVAILABLE_ACTIONS$
Reasoning Trace:
$REASONING_TRACE$

Your task is to convert the Reasoning Trace into a deeper tree-of-thinking that enumerates major uncertainties (including multi-agent cases), where each leaf maps to exactly one next action (copied verbatim from Available Actions). The evaluator will score each scenario leaf (Likelihood, Performance) versus cost (distance/communication) to select the best action.

Rules:
1. Identify the uncertain statements in the Reasoning Trace that materially affect action choice.
   - Uncertain = assumptions, guesses, speculation, or information that may be invalid due to collaborators' past actions.
   - Example: 'The apple might be in the kitchen.' or 'Bob probably already checked the cabinet.'

2. For each uncertain statement, create a binary branch and limit the maximum tree depth to 3.
   - 'True': the assumption is correct.
   - 'False': the assumption is incorrect.

3. Expand the tree in order of (i) higher uncertainty, then (ii) greater expected impact on goal success, steps, or token usage.
   - Include multi-agent third-cases (e.g., the object existed but was already taken by the collaborator and my information is stale).
   - For each branch, continue with the next uncertainty; if none remain, attach exactly one action string copied verbatim from Available Actions.

4. Ground leaf nodes using only actions from Available Actions.
   - Do not invent new objects/IDs. Use only entities present in Reasoning Trace, Progress, or Available Actions.
   - You may communicate with the collaborator via [send_message] <'...'> even if it is not listed in Available Actions. All other actions must come from Available Actions.
   - If two different scenarios lead to the same action, keep separate leaves (the evaluator still needs distinct scenarios for expected-value computation).

5. Output only the nested JSON object that represents the scenario tree (no extra text).
   - Keys = uncertain statements (or chosen actions).
   - Values = dicts with 'True' and 'False' branches OR direct action strings.
   - Each action must preserve its full format: <name> (id), copied verbatim.

Example (paired with Reasoning Trace and Available Actions):

Example Reasoning Trace:
- You are in the bathroom and holding nothing. Goal: put 3 cutlery forks into the dishwasher.
- Distances — Kitchen: 7.38m, Living room: 13.89m, Bedroom: 8.39m.
- Checking the bathroom cabinet is cheapest (3.35m) and might contain a fork.

Available Actions:
- [gocheck] <bathroomcabinet> (190) - my cost: 3.35 meters
- [goexplore] <kitchen> (11) - my cost: 7.38 meters
- [goexplore] <livingroom> (267) - my cost: 13.89 meters
- [goexplore] <bedroom> (172) - my cost: 8.39 meters
- [send_message]

```
Example Output (valid JSON):
{
 'type': 'hypothesis',
 'text': 'The <bathroomcabinet> (190) contains a <cutleryfork>',
 'reason': 'Nearby low-cost container; utensils may be stored here.',
 'True': {
   'type': 'action',
   'action': '[gocheck] <bathroomcabinet> (190) - my cost: 3.35 meters'
 },
 'False': {
   'type': 'hypothesis',
   'text': 'Bob already took a <cutleryfork> from the <bathroomcabinet> (190)',
   'reason': 'Lack of observation about collaborator's past actions; object could have been collected earlier.',
   'True': {
     'type': 'action',
     'action': '[send_message] <\'Bob, did you already take a fork from the bathroom cabinet?\'>'
   },
   'False': {
     'type': 'hypothesis',
     'text': 'Forks are in the <Kitchen> (11) (more likely than other rooms)',
     'reason': 'Utensils are commonly in the kitchen; cost to kitchen (7.38m) is lower than living room (13.89m).',
     'True': {
       'type': 'action',
       'action': '[goexplore] <kitchen> (11) - my cost: 7.38 meters'
     },
     'False': {
       'type': 'hypothesis',
       'text': 'Next best room to explore (by distance) is the <Bedroom> (172) over the <Living room> (267)',
       'reason': 'Bedroom cost (8.39m) < Living room cost (13.89m); both unexplored.',
       'True': {
         'type': 'action',
         'action': '[goexplore] <bedroom> (172) - my cost: 8.39 meters'
       },
       'False': {
         'type': 'action',
         'action': '[goexplore] <livingroom> (267) - my cost: 13.89 meters'
       }
     }
   }
 }
}
```

Figure 9: Prompts template for Composer.

**Evaluator**

I'm $AGENT_NAME$. I have received a Scenario Tree produced by the checker. Each leaf corresponds to a concrete next-action choice under a specific set of assumptions (including multi-agent cases such as collaborator-taken objects or stale information).

Tree format you may receive:
- Hypothesis node:
  {'type': 'hypothesis', 'text': '<assumption>', 'reason': '<why this is considered>', 'True': {..}, 'False': {..}}
- Action node (leaf):
  {'type': 'action', 'action': '[action_string]'}

Your task: Evaluate all actionable leaves (each leaf that contains a concrete action) and return a compact mapping that the selector can rank. For each scenario (leaf), provide:

1. Likelihood (1–5):
   - How plausible it is that this branch (set of assumptions) is true.
   - 1 = very unlikely, 5 = highly likely.

2. Gain (1–5):
  - How much this action advances/satisfies the goal under this branch.
  - 5 = directly satisfies or the most impactful next step; 3 = moderate progress; 1 = marginal.

3. CostPenalty (1–5):
  - How costly, risky, or burdensome this action is to perform (in terms of effort, time, or communication overhead).
  - Consider physical movement, communication, and resource usage
  - 1 = very low cost: communication ([send_message]), or a short move less than 3m
  - 3 = moderate cost: physical move less than 10m, or multiple small actions combined.
  - 5 = very high cost: long movement across multiple rooms (>15m), heavy resource usage, or actions likely to waste significant time/effort.

4. Action:
  - Exactly one action from Available Actions that corresponds to this scenario branch.
  - Copy the full action string verbatim (e.g., '[gograb] <apple> (36)'). If the leaf already contains a concrete action string, use that.

Constraints:
- Use only actions that appear in Available Actions (verbatim match; do not invent objects/IDs).
- You may communicate with the collaborator via [send_message] <'...'> even if it is not listed in Available Actions.
- Evaluate every leaf that contains an action. If the Scenario Tree has N actionable leaves, return N scenarios.
- Provide Likelihood, Gain, and CostPenalty as defined above. Performance will be computed by the system.
- Number scenarios in a stable order (e.g., left-to-right depth-first traversal): 'Scenario 1', 'Scenario 2', ...

Input:
Goal: $GOAL$
Progress: $PROGRESS$
Dialogue history:
$DIALOGUE_HISTORY$
Previous actions: $ACTION_HISTORY$
Available Actions:
$AVAILABLE_ACTIONS$
Scenario Tree:
$SCENARIO_TREE$

Output Format:
Return a JSON dictionary (no extra text) where:
- Keys = scenario identifiers (e.g., 'Scenario 1', 'Scenario 2', ...).
- Values = a dictionary with:
  - 'Likelihood': integer (1–5)
  - 'Gain': integer (1–5)
  - 'CostPenalty': integer (1–5)
  - 'Action': string

Example Output:
```
{
 'Scenario 1': {
   'Likelihood': 4,
   'Gain': 5,
   'CostPenalty': 3,
   'Action': '[gograb] <cutleryfork> (373)'
 },
 'Scenario 2': {
   'Likelihood': 3,
   'Gain': 4,
   'CostPenalty': 5,
   'Action': '[gocheck] <kitchencabinet> (75)'
 },
 'Scenario 3': {
   'Likelihood': 4,
   'Gain': 2,
   'CostPenalty': 1,
   'Action': '[send_message] <"Bob, did you check the kitchen?">'
 }
}
```

Figure 10: Prompts template for Evaluator.

