# OpenReview forum: "From Assumptions to Actions: Turning LLM Reasoning into Uncertainty-Aware Planning for Embodied Agents"
_ICLR.cc/2026/Conference — ICLR 2026 Poster_

### Official Review · Reviewer_yuvs · 2025-10-25

**Soundness:** 2
**Presentation:** 3
**Contribution:** 3
**Rating:** 6
**Confidence:** 4

**Summary:**

The paper proposes **PCE (Planner–Composer–Evaluator)**, a modular framework that enhances LLM agents’ ability to plan and act in multi-agent embodied environments. The **Planner** generates potential next steps, the **Composer** extracts hypotheses from plans and structures them into a decision tree, and the **Evaluator** assesses candidate actions based on expected gain and cost of paths on the decision tree. The system aims to make LLM agents more consistent and efficient by explicitly modeling future actions and uncertainty. Experiments conducted on two benchmarks, C-WAH and TDW-MAT, across multiple backbone LLMs demonstrate improvements in task success rate with decreased communication. Furthermore, human studies on PCE show that the system is more helpful and efficient.

**Strengths:**

- The **motivation for PCE is clear and well-grounded**. By designing the planning module to explicitly model upcoming steps and their corresponding confidence scores, analogous to a world model, the approach enables LLM agents to act more consistently and **avoid redundant or unproductive communications**, leading to more efficient collaboration.
- Across both C-WAH and TDW-MAT benchmarks, **PCE consistently achieves faster goal completion and higher success rates** under all three backbone LLMs (GPT-4o mini, GPT-OSS:20B, and Gemma3:4B). This demonstrates the strong performance of PCE. Meanwhile, smaller communication times make PCE more efficient when cooperating with humans.
- The authors systematically evaluate the necessity of each component and provide clear explanations of their functionalities. This detailed analysis and transparent modular design make the system **relatively easy to reproduce and adapt** to other tasks or environments.

**Weaknesses:**

- The designs of the planner, composer, and evaluator in the planning module essentially prompt LLMs to perform planning, extraction, and evaluation. Meanwhile, the evaluation metric is the score output by an LLM judge. Given that LLM scores are not very accurate, it may be challenging to accurately measure the system's stability.
- As in Tables 1 and 2, regarding the token usage metric, the result is not significant. This makes me think about whether the system is sending too many tokens at one time in the communication channel, or if the system has a higher latency in planning for the next step.
- Typos:
    - In the composer part of Fig 1., “assumtion” should be “assumption”.

**Questions:**

- Regarding Weakness 1, can the authors analyze the accuracy of the LLM-generated scores? It is important to understand their stability.
- Regarding Weakness 2, for the metrics, does the **Usage** metric measure only communication tokens or all tokens generated within the system? If it’s the former, could the authors also compare the total generated tokens with the baselines? I believe this would be a good proxy for both the latency and cost of the designed system.
- From Table 3, we observe an interesting phenomenon: removing the **planner** dramatically increases the number of communication rounds (from 1.70 to 9.52), whereas removing the **composer** decreases them. Could the authors explain why this happens?

---

> ### Author Response · Authors · 2025-11-24
>
> We are grateful for your careful review and meaningful suggestions. The following section contains our point-by-point responses, which we hope address your concerns clearly. We would be glad to continue the discussion to help resolve any remaining issues.
>
> > W1 & Q1: Accuracy of LLM-Generated Scores
>
> We analyze the reliability of LLM generated evaluation through a human expert correlation study.
>
> **1. Experimental Setup:**
>
> - **Annotators:** Four domain experts familiar with the C-WAH benchmark.
> - **Data:** We sampled execution logs from 10 C-WAH tasks. Experts received the same context and scenario trees as the Evaluator and manually assigned scores (1-5 scale) for Likelihood, Gain, and Cost. Likelihood estimates the probability that a scenario is true, Gain estimates the expected benefit of executing the leaf action if the scenario holds, and Cost estimates the expense incurred by performing that action.
>
> **2. Quantitative Results:** The following table reports the Mean Absolute Error (MAE) between Human and Evaluator scores across the three criteria.
>
> **Table: MAE of Quantitative Evaluation Results.**
> |  |  |  |
> | --- | --- | --- |
> | **Likelihood** | **Gain** | **Cost** |
> | 0.91 | 1.10 | 0.88 |
>
> Lower values indicate closer alignment between the Evaluator and human experts. Across all three criteria, the Evaluator exhibits a MAE close to 1, indicating consistently high alignment with human expert judgments despite being LLM-based.
>
> Furthermore, we provide indirect evidence supporting the stability of the Evaluator through our main experimental results (Table 1, 2). As the backbone model improves from Gemma to GPT-4o-mini, we observe consistently better planning performance, indicating that enhanced reasoning capability leads to more coherent and reliable utility estimation. This trend suggests that the Evaluator’s scores are not arbitrary outputs, but reflect structured reasoning behavior that improves with model capacity, reinforcing its reliability as a stability metric. Full details of this user study in Appendix A.10.
>
> > W2 & Q2: Does Usage Measure All Tokens and Is Total Token Cost a Reliable Proxy for Latency
>
> We clarify that Usage measures the total number of tokens generated by the entire system. This includes communication tokens and all internal tokens from the Planner Composer and Evaluator. Therefore the reported Usage already reflects total computational cost and is a valid proxy for system latency and runtime overhead.
>
> The comparable total token usage relative to baselines is the result of a deliberate allocation strategy. PCE reduces tokens spent on repetitive communication and extended episode length by completing tasks in fewer steps. These savings are reallocated to internal reasoning for higher quality decision making. As a result total token volume remains similar while success rate increases significantly.
>
> This indicates that PCE does not reduce computation by simplification but improves efficiency by shifting tokens from low value trial and error exploration to targeted reasoning. We updated Section 5 to explicitly define Usage as total token consumption across all modules to remove ambiguity and ensure fair comparison.

---

> > ### Author Response · Authors · 2025-11-24
> >
> > > Q3: Why Communication Increases without the Planner but Decreases without the Composer
> >
> > We thank the reviewer for highlighting this interesting contrast. The contrasting communication patterns in Table 3 arise from how each module regulates information seeking under uncertainty.
> >
> > **1. w/o Planner (1.70 $\rightarrow$ 9.52 Comm):** Ambiguity leading to Dependency.
> >
> > Without the Planner to generate a proactive initial strategy, the agent lacks a coherent direction. Faced with unstructured context, the agent struggles to form independent physical plans and defaults to reactive behavior. Consequently, it relies heavily on communication (asking for information or instructions) as a fallback mechanism to resolve its confusion, leading to a sharp increase in dialogue.
> >
> > **2. w/o Composer (1.70 $\rightarrow$ 0.26 Comm):** Overconfidence leading to Rashness.
> >
> > Without the Composer, the agent operates only on the Planner’s single trajectory reasoning. As discussed in the paper (Section 4.2), raw LLM traces tend to be "single-minded" or overconfident—often assuming a likely state is true without exploring alternatives. Lacking the mechanism to explicitly ask "What if this assumption is wrong?", the agent fails to recognize the value of information gathering. It thus skips communication and prematurely commits to physical actions, resulting in near-zero communication and suboptimal performance.
> >
> > These results show that the Planner reduces unnecessary dependency by providing goal oriented structure, while the Composer promotes justified communication by exposing latent uncertainty. Balanced interaction between the two enables stable and efficient communication behavior.
> >
> > > W3: Typos
> >
> > We sincerely thank the reviewer for the careful observation. We have corrected the typo ("assumtion" $\rightarrow$ "assumption") in Figure 1 of the revised manuscript.

---

> > > ### Comment · Reviewer_yuvs · 2025-11-26
> > >
> > > Thank you for the response. I would like to keep the current score and am inclined to accept the paper.

---

### Official Review · Reviewer_SbnL · 2025-10-28

**Soundness:** 2
**Presentation:** 2
**Contribution:** 1
**Rating:** 2
**Confidence:** 4

**Summary:**

This paper proposes PCE (Planner-Composer-Evaluator), a framework for uncertainty-aware planning in LLM-based embodied agents. The key idea is to extract implicit assumptions from LLM reasoning traces and structure them into a decision tree, enabling effective multi-agent coordination without heavy communication. However, the technical contribution is limited and differentiation from similar work is unclear.

**Strengths:**

- The paper effectively identifies the communication overhead problem.
- User study results showing that PCE produces more efficient communication patterns demonstrate practical value.
- Consistent improvements across GPT-4o mini, GPT-OSS:20B, and Gemma3:4B suggest broad applicability.

**Weaknesses:**

- The distinction between the proposed methodology and existing multi-agent task planning techniques remains unclear. In particular, the paper needs to articulate clear differences from approaches like ProAgent, CoELA, REVECA, and CaPo, which also perform tasks through multi-agent communication.
- Furthermore, while the main contributions are presented as the Planner-Composer-Evaluator structure and the decision tree-based techniques in the Composer and Evaluator, these appear to be applications of existing methods rather than novel contributions.
- In the experimental section, the performance degradation when removing the Composer appears minimal. This raises questions: if the Planner-Evaluator structure alone achieves higher performance than baselines that utilize collaborative agents, what accounts for this superiority? The analysis lacks clarity on why performance exceeds baselines even without the Composer, and where specifically the Composer contributes to performance improvements. Such analysis should be included to better understand the contribution of each component.

**Questions:**

- What is the recovery method when assumptions at decision tree nodes are incorrect?
- How does PCE perform in environments where communication is completely blocked?
- How do you evaluate and ensure the quality of assumption extraction?

---

> ### Author Response · Authors · 2025-11-24
>
> Thank you for the valuable and well-articulated comments. We provide our point-by-point replies below, and we hope these responses adequately address your questions. We also look forward to engaging in additional discussion to improve the work.
>
> > W1 & W2: How PCE Differs from Existing Multi-Agent Planning and Why It Is Novel
>
> We believe the reviewer’s concern reflects a fundamental distinction that we did not articulate with sufficient clarity in the original manuscript. We therefore take this opportunity to clarify this distinction, as existing multi-agent planners and PCE operate at fundamentally different levels of the decision problem.
>
> Prior methods such as ProAgent, CoELA, REVECA and CaPo focus on optimizing coordination once the problem structure is assumed to be known. Their primary challenge is how multiple agents exchange information efficiently and synchronize plans, typically treating communication as a required protocol to reduce uncertainty. In these systems, uncertainty is something to be eliminated through more dialogue.
>
> PCE (our method) operates at a fundamentally different level. It does not ask how agents should communicate, but whether they should communicate at all. The core novelty is that PCE explicitly models uncertainty as a first-class object of reasoning and introduces a structured mechanism to reason about belief uncertainty before any coordination occurs.
>
> The Planner produces a free-form reasoning trace as most LLM systems do. However, instead of directly acting on this trace, the Composer transforms it into a structured hypothesis space of latent assumptions. These assumptions represent alternative world states that may or may not be true. The Evaluator then performs utility-based comparison across these hypothetical worlds, allowing the agent to decide whether communication, physical exploration, or continued independent action is most rational.
>
> This shift from planning actions to reasoning over assumptions is not an incremental extension of prior work. It introduces meta-reasoning over the agent’s own belief state, aligning with recent AI discourse on uncertainty-aware decision making, belief modeling, and belief-based reasoning in DEC-POMDP settings. In simple terms, prior methods optimize plans, while PCE optimizes the process of deciding which beliefs to trust.
>
> Therefore, the novelty of PCE is not in using a tree, but in redefining what the tree represents: a structured space of belief uncertainty rather than a space of action sequences or communication strategies. This enables selective communication, principled self-correction, and stability under partial observability in a way that existing communication-centric planners do not address.
>
> In this sense, PCE should be viewed not as another coordination strategy, but as a new reasoning layer that governs when coordination itself is necessary.
>
> > W3: Why w/o Composer Still Outperforms Baselines and What the Composer Actually Adds
>
> The reviewer correctly notes that removing the Composer causes only moderate performance degradation. This reveals two distinct effects.
>
> **1. Why "w/o Composer" > Baselines? (The Cost of Forced Communication):**
> The Planner-Evaluator structure (w/o Composer) alone outperforms communication heavy baselines because it explicitly evaluates action utility and cost. Existing methods (e.g., CoELA) trigger communication whenever uncertainty arises, which leads to repeated dialogue and replanning cycles. In contrast, even without the Composer, the Evaluator suppresses unnecessary communication and prioritizes cost effective physical actions. This selective restraint explains why w/o Composer still surpasses collaborative baselines.
>
> **2. Specific Contribution of the Composer (Handling "Critical" Uncertainty):**
>
> The Composer contributes by enabling scenario level reasoning under critical uncertainty. Without it, the Planner tends to follow a single high confidence trajectory, making it vulnerable to hallucinated or overconfident decisions. The Composer introduces alternative assumption branches and allows the Evaluator to score competing scenarios rather than isolated actions. The improvement from (46.82 $\rightarrow$ 42.76 steps) reflects reduced failure cascades such as committing to an incorrect search path that would otherwise require costly recovery.
>
> In short, w/o Composer wins by avoiding unnecessary coordination, while full PCE wins by preventing strategically wrong decisions through structured uncertainty modeling. Qualitative examples are provided in Appendix A.7.

---

> ### Author Response · Authors · 2025-11-24
>
> > Q1: How PCE Recovers When Decision Tree Assumptions Are Incorrect
>
> When assumptions at decision tree nodes are incorrect, PCE handles this through likelihood-based rejection (Soft Filtering) and closed-loop replanning.
>
> **1. Likelihood-based rejection via Evaluator**: Incorrect assumptions are suppressed before execution. The Evaluator assigns low likelihood to scenarios that contradict current observations or prior knowledge, which directly reduces their expected utility score. This prevents most faulty assumptions from being selected as actions.
>
> **2. Closed-Loop Replanning (Recovery):** If an incorrect assumption is executed and proven false through interaction, the system recovers via feedback-driven replanning. The failed outcome is stored in memory and incorporated into the next planning cycle. The Planner and Composer then regenerate the decision tree conditioned on the updated observation, naturally discarding the invalid hypothesis and selecting an alternative action.
>
> As a result, incorrect assumptions do not lead to permanent failure but rather serve as information-gathering steps that refine the agent's belief state.
>
> > Q2: How PCE Performs When Communication Is Completely Blocked
>
> PCE continues to operate effectively even when communication is fully unavailable.
>
> When communication is blocked, PCE simply excludes send_message from the candidate action set. The Composer constructs decision trees based solely on physical actions, and the Evaluator selects actions using the same utility based reasoning. Since PCE fundamentally optimizes expected utility over belief uncertainty rather than relying on explicit dialogue, its planning logic remains stable.
>
> Empirically, this scenario is evaluated in Table 8 of Appendix A.5 under the Phy-act only setting. The results are as follows:
>
> - **PCE (with Comm):** 42.76 steps
> - **PCE (Comm blocked / Phy-act only):** 45.38 steps
>
> These results demonstrates that communication improves coordination efficiency but is not structurally required. In the absence of communication, PCE compensates through more deliberate physical exploration driven by belief aware utility optimization, preserving robust decision making in isolated environments.
>
> > Q3: How We Evaluate and Ensure the Quality of Assumption Extraction
>
> We agree that ensuring the quality of assumption extraction is critical for the framework’s integrity. To rigorously assess this component, we conducted a quantitative evaluation to verify whether the assumptions extracted or newly generated by the Composer are free from hallucinations or errors.
>
> Preparing and analyzing these evaluations requires substantial time. We are actively finalizing the results and will provide them in an additional comment as soon as possible.

---

> > ### Author Response · Authors · 2025-11-26
> >
> > > Q3: How We Evaluate and Ensure the Quality of Assumption Extraction
> >
> > To quantitatively assess the efficacy of the Composer’s assumption extraction at the node level, we designed an experimental evaluation using execution logs from 10 tasks in the C-WAH benchmark. Given that the validity of assumptions is inherently semantic and context-dependent, algorithmic metrics or environment rewards alone are insufficient for reliable validation. Therefore, we rely on human experts as the primary reference to judge whether an assumption genuinely reflects the Planner’s intent or constitutes a reasonable hypothesis in ambiguous contexts.
> >
> > **1. Data Construction**
> >
> > We modified the original Composer prompt to enable self-tagging for each assumption node. The system classifies each assumption as either (a) Extracted, indicating it was derived directly from the Planner's reasoning trace, or (b) Generated, indicating it was synthesized by the Composer as a novel hypothesis not explicitly stated in the trace.
> >
> > Furthermore, to evaluate the accuracy of assumption extraction relative to the quality of the Planner's reasoning trace, four domain experts categorized each instance into three difficulty levels (Easy, Medium, Hard) based on identical inputs (environmental state, goal, and Planner trace). Easy cases contain assumptions explicitly stated with minimal noise. Medium cases involve assumptions that are inferable but embedded in ambiguous or cluttered reasoning. Hard cases correspond to vague traces where the critical assumption is not directly mentioned, requiring implicit inference or hypothesis generation.
> >
> > This process established a stress-testing dataset encompassing a spectrum of scenarios, ranging from clear reasoning traces to highly ambiguous ones.
> >
> > **2. Evaluation Methodology**
> >
> > Each assumption generated by the Composer was subsequently evaluated by three additional domain experts to determine its validity based on the following criteria:
> >
> > - **For assumptions labeled as 'Extracted':** The assumption must be explicitly supported by the reasoning trace.
> > - **For assumptions labeled as 'Generated':** While the assumption does not appear in the reasoning trace, it must form a contextually plausible hypothesis given the environment, goal, and agent memory.
> >
> > The final validity label for each assumption was determined via majority voting among the three experts. We then analyzed the correctness of the individual assumptions generated by the Composer across the distinct difficulty intervals.
> >
> > **3. Results and Interpretation**
> >
> > | Easy | Medium | Hard | Overall |
> > | --- | --- | --- | --- |
> > | 84.3% | 77.8% | 76.7% | 80.6% |
> >
> > These results demonstrate that the Composer consistently identifies appropriate assumptions and generates necessary additional hypotheses across all difficulty levels. The system exhibits high robustness, maintaining performance even when the reasoning trace is ambiguous. Achieving an average assumption validity of 80.6%, the Composer demonstrates a significant level of reliability.  We include the full details of these results in Appendix A.11.

---

> > > ### Comment · Reviewer_SbnL · 2025-11-27
> > >
> > > Thank you for your detailed response. I have an additional question.
> > >
> > > 1. The authors claim that PCE introduces "meta-reasoning over the agent's own belief state," distinguishing it from prior work that merely optimizes plans. However, the Composer's assumption extraction seems to rely primarily on LLM prompting (Figure 9) rather than any formal belief representation. Could the authors clarify how this "meta-reasoning" rather than simply structured prompt engineering? Specifically, how does the decision tree fundamentally differ from Tree-of-Thoughts, which also branches on reasoning steps?
> > >
> > > 2. The authors state that the Composer handles critical uncertainty and prevents failure cascades. However, the paper does not provide a formal definition of what constitutes critical/non-critical uncertainty. How does the Composer's local ranking policy determine which assumptions most reduce uncertainty and most strongly influence subsequent action choice? Is this learned or just using prompt? If the Composer's assumption ranking relies solely on prompting, how can you guarantee the quality and consistency of its choices?
> > >
> > > 3. It seems to both the Composer and Evaluator rely on the same LLM backbone (e.g., GPT-4o mini). if the LLM generates a hallucinated assumption in the Composer, might it not also assign high likelihood to that same assumption in the Evaluator?

---

> > > > ### Author Response · Authors · 2025-11-29
> > > >
> > > > > Q1 - The authors claim that PCE introduces "meta-reasoning over the agent's own belief state," distinguishing it from prior work that merely optimizes plans. However, the Composer's assumption extraction seems to rely primarily on LLM prompting (Figure 9) rather than any formal belief representation. Could the authors clarify how this "meta-reasoning" rather than simply structured prompt engineering? Specifically, how does the decision tree fundamentally differ from Tree-of-Thoughts, which also branches on reasoning steps?
> > > >
> > > > Thank you for the insightful follow-up. We hope the clarification below helps resolve your concerns.  We appreciate the opportunity to clarify the meaning of “meta-reasoning over the agent’s own belief state” more precisely.
> > > >
> > > > **What do we mean by "the agent’s own belief state"?**
> > > >
> > > > In our framework, the "belief state" is the structured set of assumptions that the Planner implicitly encodes in its reasoning trace and memory. These assumptions capture what the agent currently takes to be true, uncertain, or likely about the world (for example, "the apple is probably in the kitchen," "the collaborator has already checked the bedroom"). They are not a probabilistic belief in the POMDP sense, but they serve as a qualitative proxy for world uncertainty from the agent’s internal perspective.
> > > >
> > > > The Composer's target is precisely this assumption space: it does not operate on the raw environment state, but on the Planner’s own assumptions about that state. This is what we mean by "the agent’s own belief state."
> > > >
> > > > **Why does this constitute meta-reasoning rather than just prompt engineering?**
> > > >
> > > > Standard LLM planners use prompting at the object level: the model is asked directly, "What should I do next?" and the resulting Chain-of-Thought trace is used to justify an action. PCE deliberately introduces a second, structurally distinct layer on top of this.
> > > >
> > > > - The Planner still operates at the task level, producing an action proposal and a reasoning trace.
> > > > - The Composer treats that reasoning trace as an object. It asks, in effect, "What assumptions is this plan resting on, and in what alternative ways could the world plausibly differ?" It then builds a scenario tree whose nodes are explicit assumptions and whose root–to–leaf paths are alternative belief configurations.
> > > > - The Evaluator does not re-plan directly in the environment. Instead, it evaluates each belief-action root-to-leaf pair $(S, a)$ in terms of Likelihood, Gain, and Cost, then chooses the action under the belief configuration with the highest expected utility.
> > > >
> > > > Thus, the final decision is made after restructuring and re-evaluating the Planner’s own assumptions, rather than directly from the Planner’s trace. In this sense, the upper layer of PCE reasons about the agent’s internal belief candidates, not about the environment itself. The prompts are an implementation vehicle, but the computation they instantiate is a two-level architecture: first-order planning below, and belief-level arbitration above. That belief-level arbitration is what we refer to as meta-reasoning.

---

> > > > > ### Author Response · Authors · 2025-11-29
> > > > >
> > > > > **How does this differ from Tree-of-Thoughts (ToT)?**
> > > > >
> > > > > ToT and PCE both employ tree structures, but they operate on fundamentally different objects.
> > > > > ToT’s nodes are candidate thought states on the way to a solution. It branches over alternative reasoning steps or partial solutions in order to search for a better plan.
> > > > > PCE’s nodes are explicit assumptions about the world or collaborators, and each path represents a distinct hypothesised world configuration.
> > > > > For example, consider a case where the agent must find an apple. A ToT formulation might branch as follows:
> > > > >
> > > > > - Thought A: "Search the kitchen first because apples are usually stored there."
> > > > > - Thought B: "Ask collaborator before moving."
> > > > > - Thought C: "Search bedroom first as a fallback."
> > > > > These are alternative reasoning strategies.
> > > > >
> > > > > In PCE, the branching instead reflects alternative world hypotheses:
> > > > >
> > > > > - Assumption 1: "The apple is in the kitchen."
> > > > > - Assumption 2: "The apple has already been taken by the collaborator."
> > > > > - Assumption 3: "The apple is still unobserved and may be in the bedroom."
> > > > >
> > > > > Each root-to-leaf path corresponds to a distinct possible world configuration, and the action at the leaf is chosen conditional on that configuration. The tree, therefore, represents not how the agent reasons, but what the agent believes about the world.
> > > > >
> > > > > This distinction is crucial.
> > > > >
> > > > > - Tree-of-Thoughts asks: "How should I reason better to achieve my goal?"
> > > > > - PCE asks: "Which version of the world am I implicitly assuming, and which assumption should I act on?"
> > > > >
> > > > > This shifts the planning paradigm from searching over reasoning trajectories to searching over belief configurations. The agent is no longer optimizing a chain of thoughts; it is arbitrating between competing interpretations of reality, explicitly scoring them by likelihood, expected gain, and cost.
> > > > >
> > > > > The tree in PCE is not a mechanism for improving reasoning fluency, but a structured representation of alternative possible worlds extracted from the agent's own internal assumptions.

---

> > > > > > ### Author Response · Authors · 2025-11-29
> > > > > >
> > > > > > > Q2 - The authors state that the Composer handles critical uncertainty and prevents failure cascades. However, the paper does not provide a formal definition of what constitutes critical/non-critical uncertainty. How does the Composer's local ranking policy determine which assumptions most reduce uncertainty and most strongly influence subsequent action choice? Is this learned or just using prompt? If the Composer's assumption ranking relies solely on prompting, how can you guarantee the quality and consistency of its choices?
> > > > > >
> > > > > > **Definition of critical uncertainty.**
> > > > > >
> > > > > > In PCE, critical uncertainty is defined operationally rather than symbolically. An assumption is considered critical when its truth value exhibits high decision sensitivity: resolving it leads to a substantially different ranking of candidate actions in terms of likelihood, expected gain, and cost. In other words, critical assumptions are those whose True/False branches generate meaningfully different utility landscapes, thereby altering which action is rationally preferred. This pragmatic definition allows the agent to identify impactful sources of uncertainty even in semantically complex domains.
> > > > > >
> > > > > >
> > > > > >
> > > > > > **Why we do not impose a symbolic definition.**
> > > > > >
> > > > > > Classical belief-space planning relies on well-defined state variables. In our target setting, however, the uncertainties arise from unstructured visual content, collaborator intentions, dynamic object states, and implicit commonsense. Constructing a handcrafted symbolic definition of uncertainty categories would not scale to such semantic diversity. Instead, we treat assumptions as language-grounded belief variables extracted from the Planner’s reasoning trace. Their criticality is revealed by their impact on downstream decision utility rather than by rule-based classification. This aligns with emerging paradigms in LLM-based planning, where belief quality is evaluated by decision consequences rather than symbolic completeness.
> > > > > >
> > > > > > **How the Composer prioritizes assumptions.**
> > > > > >
> > > > > > The Composer’s local ranking policy is designed to effectively partition the hypothesis space, operating by selecting assumptions that most reduce uncertainty and most strongly influence subsequent action choice. Concretely, an assumption is ranked higher when its True/False outcomes significantly narrow the set of feasible subsequent assumptions and constrain the valid set of leaf actions. This acts as a divide-and-conquer criterion over the hypothesis space: each branch expansion reduces the uncertainty that remains reachable within the allowed tree depth. Because the branching immediately restricts both assumption candidates and action candidates, the Composer explores a smaller, more coherent search space at every step.
> > > > > >
> > > > > > This structure provides an additional benefit: reducing the search horizon lowers the difficulty presented to the LLM, thereby decreasing the likelihood of hallucinated or logically incompatible assumptions. Similarly, scenarios generated under sharply constrained assumptions admit only a small set of valid actions, reducing the probability of assigning sub-optimal or contextually inconsistent actions.

---

> > > > > > > ### Author Response · Authors · 2025-11-29
> > > > > > >
> > > > > > > **Prompt-based policy and quality guarantees.**
> > > > > > >
> > > > > > > The Composer’s ranking policy is implemented via prompting rather than learned parameters. Quality and consistency are ensured through two complementary mechanisms.
> > > > > > >
> > > > > > > First, the design itself structurally reduces opportunities for hallucination: at each step the Composer conditions only on assumptions consistent with the current combination of assumptions and prunes incompatible branches, forcing the LLM to operate within a narrowed and progressively more coherent belief space.
> > > > > > >
> > > > > > > Second, PCE incorporates an Evaluator layer that converts flawed assumptions into lower expected utility, serving as a probabilistic safety mechanism that prevents Composer errors from causing failure cascades. This issue is addressed in detail in our response to the reviewer’s third follow-up question.
> > > > > > >
> > > > > > > Additionally, we conducted a dedicated expert evaluation of the decision-tree outputs. Four domain experts independently reconstructed decision trees from identical Planner inputs, and both human- and Composer-generated trees (excluding each evaluator’s own) were blindly cross-reviewed on a 7-point Likert scale across five dimensions.
> > > > > > >
> > > > > > > | Metric | Composer | Human | Evaluation Focus |
> > > > > > > | --- | --- | --- | --- |
> > > > > > > | Q1 Extraction Accuracy | 6.27 | 6.64 | Does the extracted assumption actually exist in the trace? |
> > > > > > > | Q2 Generation Capability | 6.39 | 6.73 | Are new assumptions generated appropriately when the trace is vague? |
> > > > > > > | Q3 Ranking Logic | 5.83 | 6.23 | Are nodes ordered by uncertainty reduction and influence on action choice? |
> > > > > > > | Q4 Logical Consistency | 6.18 | 6.55 | Are there contradictions in the decision tree? |
> > > > > > > | Q5 Action Appropriateness | 5.98 | 6.15 | Do leaf actions match the scenario path? |
> > > > > > >
> > > > > > > The Composer achieves performance close to human experts across all metrics. Notably, strong scores in Q2 and Q4 indicate that the Composer maintains coherence even when generating new assumptions beyond the input trace. Q3 further shows that the ranking policy effectively identifies assumptions that reduce uncertainty and influence downstream actions. Q5 confirms that assigned leaf actions correctly correspond to the scenario paths.
> > > > > > >
> > > > > > > **Summary.**
> > > > > > >
> > > > > > > Critical uncertainty in PCE is defined by decision sensitivity, not symbolic rules. This definition enables scalable belief modeling in semantically rich environments and provides a principled criterion for assumption ranking. The Composer’s prompt-based ranking, combined with uncertainty-reducing branching, reduces hallucination risk and improves consistency. Expert evaluation and stress-test results demonstrate that the Composer’s decision-tree quality is near–human-expert level, and the Evaluator further safeguards against residual Composer errors, ensuring stable and reliable performance.

---

> > > > > > > > ### Author Response · Authors · 2025-11-29
> > > > > > > >
> > > > > > > > > Q3 - It seems to both the Composer and Evaluator rely on the same LLM backbone (e.g., GPT-4o mini). if the LLM generates a hallucinated assumption in the Composer, might it not also assign high likelihood to that same assumption in the Evaluator?
> > > > > > > >
> > > > > > > > As noted by the reviewer, while the Composer and Evaluator share the same LLM backbone, they operate under distinct conditions with different inputs and objectives. Specifically, the Composer generates assumptions by interpreting and expanding upon the Planner’s free-form reasoning trace. In contrast, the Evaluator does not receive this reasoning trace; instead, it compares the proposed decision tree against a context input composed of objective facts, such as observations, memory, and message history. Consequently, the Evaluator calculates likelihood based on the logical consistency between the scenario's assumptions and the given context. This means that even assumptions generated by the Composer (sharing the same backbone) will struggle to achieve a high likelihood score if they conflict with the objective context.
> > > > > > > >
> > > > > > > > This is further evidenced by our additional experiments. In the assumption-level stress test described previously, we annotated the validity of each individual assumption. Using this data, we conducted an additional analysis to examine how the Evaluator assesses scenarios containing invalid assumptions.
> > > > > > > >
> > > > > > > > To this end, we categorized the scenarios into two groups:
> > > > > > > >
> > > > > > > > 1. Scenarios consisting entirely of valid assumptions.
> > > > > > > >
> > > > > > > > 2. Scenarios containing at least one invalid assumption.
> > > > > > > >
> > > > > > > > We then calculated the average likelihood and utility scores assigned by the Evaluator for each group. The results are presented below:
> > > > > > > >
> > > > > > > > | **Scenario Type** | **Likelihood** | **Utility Score** |
> > > > > > > > | --- | --- | --- |
> > > > > > > > | **Valid assumptions only** | 3.28 | 3.63 |
> > > > > > > > | **Contains invalid assumptions** | 2.85 | 2.51 |
> > > > > > > >
> > > > > > > > Scenarios containing invalid assumptions received lower scores in both Likelihood and Utility.
> > > > > > > >
> > > > > > > > The decrease in Likelihood scores indicates that the Evaluator successfully recognizes inconsistencies regarding the assumptions. More importantly, the decline in Utility scores implies that the Evaluator not only identifies flawed assumptions but also effectively filters out the final actions derived from those incorrect assumptions.
> > > > > > > >
> > > > > > > > Further details regarding this experiment can be found in Appendix A.11.

---

### Official Review · Reviewer_FyaB · 2025-10-28

**Soundness:** 3
**Presentation:** 3
**Contribution:** 3
**Rating:** 6
**Confidence:** 4

**Summary:**

This paper identifies a key challenge for LLM-based embodied agents in partially observable, multi-agent settings: the heavy reliance on inter-agent communication to resolve uncertainty, which incurs significant token, time, and human workflow costs.

To address this, the authors propose PCE, a Planner-Composer-Evaluator framework. The core idea is to leverage the implicit, fragmented assumptions that LLMs generate in their reasoning traces.

- The **Planner** produces an initial action and its reasoning trace.

- The **Composer** extracts these latent assumptions and structures them into an explicit decision tree (a "scenario tree"). Internal nodes represent assumptions (with True/False branches), and leaves represent the final action to be taken under that scenario.

- The **Evaluator** scores each root-to-leaf path (scenario) based on its estimated likelihood (L(S)), conditional gain (G(a)), and execution cost (C(a)) .

This allows the agent to make a rational, uncertainty-aware choice—including whether to communicate or take a physical action—by selecting the action from the scenario with the highest final utility score, U(S,a)=E[gain]−λC(a). Experiments on two multi-agent benchmarks (C-WAH and TDW-MAT) with three different LLM backbones show that PCE outperforms communication-centric baselines in success rate and efficiency. A user study also suggests that PCE's communication patterns are perceived by humans as more efficient and trustworthy.

**Strengths:**

- **Originality and Significance**: The paper's core contribution is novel and insightful. Instead of simply using an LLM's reasoning trace (like Chain-of-Thought), the PCE framework performs meta-reasoning on the trace itself. The idea of "turning LLM reasoning into... planning"  by extracting, structuring, and formally evaluating latent assumptions is a clever way to operationalize the implicit knowledge within LLMs for decision-making under uncertainty.

- **Problem Formulation**: The work addresses a well-defined and critical problem. As LLM agents become more capable, their reliance on frequent communication becomes a bottleneck, especially when humans are in the loop. This paper offers a principled alternative to naive, communication-heavy strategies.

- **Methodological Clarity**: The proposed PCE framework is modular, logical, and well-explained. The three-stage pipeline is intuitive, and the Evaluator's scoring function provides a principled mechanism for balancing scenario likelihood, potential gain, and the distinct costs of physical vs. communicative actions.

- **Thorough Empirical Evaluation**: The experimental validation is a significant strength.

  * __Generality__: The method is tested on three diverse LLM backbones (including commercial and open-source models) across two challenging benchmarks , consistently outperforming four representative baselines.

  * __Ablation Studies__: The ablations are comprehensive. The component analysis (Table 3) successfully demonstrates that each part of the PCE pipeline is necessary for good performance .

   * __Scaling Analysis__: The "LLM Scaling" study (Figure 3) provides compelling evidence that the performance gains are attributable to the PCE framework itself, not just to using a larger model. It shows that while scaling models (e.g., Gemma3 4B → 27B) improves a "Planner only" baseline, PCE reaps greater benefits, widening the performance gap .

   * __User Study__: The inclusion of a user study  is commendable. It directly validates the paper's central hypothesis: that reducing communication intelligently leads to a human-agent collaboration that is not only more efficient but also perceived as more trustworthy and useful.

**Weaknesses:**

- **Scalability of the "Multi-Agent" Claim**: The experiments are exclusively conducted in two-agent settings. While technically "multi-agent," this does not sufficiently support the paper's broader claims of solving uncertainty in "multi-agent... environments". The complexity of tracking collaborator intentions and partial observations scales combinatorially with the number of agents. It is unclear how the Composer's decision tree and the Evaluator's scoring would handle branching on assumptions about n−1 other agents without becoming intractable.

- **The Composer**: The entire framework's effectiveness hinges on the Composer module. This module is tasked with complex, non-trivial reasoning steps: (1) semantically identifying the most critical uncertainties from a free-text reasoning trace , (2) ranking them by abstract criteria like "influence" , and (3) proposing new atomic assumptions from scratch when needed. The paper states this is approximated using "LLMs' commonsense reasoning"  and provides a prompt (Figure 9), but this sweeps a massive amount of complexity under the rug. The paper offers no analysis of the Composer's reliability. If the Composer fails to extract the key assumption or hallucinates an irrelevant new one, the entire decision tree is built on a faulty foundation, and the Evaluator's "principled" scoring becomes meaningless. The ablation in Table 3 only shows that no Composer is bad, not that the current Composer is robust.


- **Potentially Overstated Scaling Claims**: In the scaling ablation (Figure 3), the performance improvement (i.e., the slope of the line) for "Planner only" appears quite similar to that of "PCE." For example, in Figure 3(b), both methods see a drop of ~9-10 steps when moving from "Low" to "High" reasoning. The paper's claim that PCE "amplifies the benefits of scaling"  seems slightly overstated; the data suggests the benefit of the PCE framework is largely additive—PCE starts at a better baseline, and that baseline advantage is maintained or slightly widened as the model scales.


- **Missing Related Work**: The related work section  focuses primarily on communication-centric methods (like ProAgent, CoELA, etc.). However, it seems to overlook other recent lines of work on long-horizon LLM planning in partially observable environments that do not rely on heavy communication. For example, [1].

[1] Nayak, Siddharth, et al. "LLaMAR: Long-Horizon Planning for Multi-Agent Robots in Partially Observable Environments." arXiv preprint arXiv:2407.10031 (2024).

**Questions:**

- **On the Composer's Reliability (W2)**: The Composer's ability to correctly identify assumptions and generate new ones is critical. How robust is this process? What happens if the Planner's reasoning trace is vague or does not contain an obvious, extractable assumption? Is there any quantitative analysis of the Composer's "hit rate" for identifying the correct critical uncertainty?

- **On Scalability (W1)**: Could you elaborate on how you expect the PCE framework, particularly the Composer's tree generation, to scale to n>2 agents? Would the tree depth D=3  be sufficient to model the compounded uncertainties from multiple collaborators?

- **Clarification of Figure 2**: The visualization in Figure 2(c) is slightly confusing. It highlights a path corresponding to nodes 1(False) → 5(False) → 4 as the "best scenario." To clarify: at the time of decision-making, the truth values of the assumptions are unknown. Does this highlighted path simply represent the leaf node (action [gocheck] cabinet) that received the highest utility score U from the Evaluator, and the path shown is the scenario (i.e., the set of assumptions) under which that action is optimal?

- **Clarification of the Communication Mechanism**: The user study strongly supports that PCE's communication is more efficient. To confirm my understanding: is communication reduced simply because the [send_message] action is treated as just another potential leaf in the decision tree, which must then "win" the U(S,a) competition against all physical actions? This seems to be the implicit gating mechanism, and it's elegant, but I want to ensure I'm not missing a more explicit component. If that is the case, its best to explicitly mention it in the paper.

---

> ### Author Response · Authors · 2025-11-24
>
> We sincerely appreciate your insightful and detailed comments. In the following, we present our point-by-point response and hope our clarifications resolve the issues you raised. We welcome any continued dialogue that can further strengthen the paper.
>
> > W1 & Q2: Scalability of the Multi-Agent Claim
>
> We address the reviewer’s concern that PCE may become intractable as the number of agents increases.
>
> **1. Complexity Is Trace-Driven, Not Agent-Driven:**
>
> The Composer does not extract uncertainty from the full joint state of all agents (which would indeed be combinatorial). Instead, it extracts uncertainty solely from the Planner's reasoning trace, observation, and memory. Since the Planner (an LLM) inherently filters out irrelevant details and focuses on salient uncertainties, the Composer’s search space scales with the complexity of these uncertainties, rather than linearly with the number of agents ($N$).
>
> **2. Semantic Aggregation Phenomenon (Handling $N > 2$):**
>
> Regarding the specific concern about branching on $N-1$ agents, we observed that LLMs naturally employ **semantic aggregation of assumptions** to handle multiple collaborators. Instead of generating separate assumptions for each agent (e.g., "Did Agent A check? Did Agent B check?"), the Planner abstracts them into collective states (e.g., "Has *any* collaborator checked the kitchen?"). This abstraction allows a fixed tree depth of $D=3$ to remain sufficient for modeling “Self vs. Others” uncertainties even as $N$ increases.
>
> **3. Empirical Validation ($N=3, 4$):** To validate this empirically, we conducted additional experiments in C-WAH with 3 and 4 agents, using the same $D=3$.
>
> | Number of Agents (N) | Total Steps (↓) |
> |---------------|------------------|
> | *N = 2*       | 42.76            |
> | *N = 3*       | 34.60            |
> | *N = 4*       | 28.50        |
>
> Performance improves with more agents, indicating that the agents successfully coordinated to divide the labor without getting overwhelmed by planning complexity. These results confirm that PCE scales robustly to small-to-medium agent teams. We acknowledge that extreme-scale coordination remains future work and discuss this limitation accordingly. We have included these results in Appendix A.9 of the revised paper.
>
> > W2 & Q1: Composer's Reliability
>
> We agree that the Composer is the core of our framework and its reliability needs rigorous verification. To address the reviewer’s concern regarding the extraction accuracy of assumptions from the Planner’s reasoning trace, generate new assumptions if needed, and effectively ranking critical assumptions, we conducted a quantitative evaluation on whether the assumptions extracted or generated by the Composer are free from hallucinations or errors, as well as an evaluation comparing Composer-generated decision trees against human-expert-generated decision trees.
>
> Preparing and analyzing these experiments has required substantial time. We will finalize them as soon as possible and include the results in an additional comment shortly.
>
> > W3: Revise the PCE’s Scaling Advantage (Avoid Overstate)
>
> We thank the reviewer for the careful analysis. We agree that the term *“amplifies”* may overstate the observed trend and have revised it accordingly.
> Figure 3 shows that while both "Planner only" and PCE benefit from increased reasoning capacity, PCE consistently achieves a stronger performance baseline that is maintained or slightly widened as scale increases. This indicates that PCE does not merely benefit from scaling, but systematically converts increased reasoning capacity into more effective uncertainty-aware decisions.
>
> We therefore revis our claim (Abstract, Related Work, and Section 5.2) to reflect that PCE provides a *stable and robust scaling advantage*, rather than a diverging one, and update the manuscript to use more precise phrasing such as "consistently enhances performance across all scales."

---

> ### Author Response · Authors · 2025-11-24
>
> > W4: Missing Related Work
>
> We thank the reviewer for bringing this important work to our attention. We agree that LLaMAR [1] is a highly relevant study addressing long-horizon planning in partially observable environments without relying on heavy communication.
>
> In the revised manuscript, we have expanded the Related Work section to incorporate this line of research. We also clarify how PCE complements and differs from these approaches. While LLaMAR focuses on planning within a centralized control setting, PCE directly addresses the cooperative challenges that arise in decentralized control environments. Specifically, by explicitly extracting, structuring, and evaluating latent assumptions, PCE performs meta-reasoning that interprets uncertainty and determines when communication or physical action is necessary. This addition provides a more balanced view of the field, extending beyond communication-centric approaches.
>
> ---
>
> References:
> [1] Nayak, Siddharth, et al. "LLaMAR: Long-Horizon Planning for Multi-Agent Robots in Partially Observable Environments." arXiv preprint arXiv:2407.10031 (2024).
>
> > Q3: Clarification of Figure 2
>
> We appreciate the reviewer for the careful clarification request. The interpretation is correct.
>
> At decision-making stage, the truth values of assumptions are unknown. The highlighted path in Figure 2(c) denotes the **scenario (assumption chain)** whose corresponding leaf action received the highest expected utility score U from the Evaluator. In other words, the path visualizes the assumption context under which the selected action (4.[gocheck]) is optimal.
>
> To remove ambiguity, we revised the caption to explicitly state: *" The Evaluator scores each path; The highlighted path indicates the scenario whose leaf node achieves the maximum expected utility U, determining the agent’s final selected action.”*
>
> > Q4: Clarification of the Communication Mechanism
>
> We confirm the reviewer’s understanding: PCE reduces communication because [send_message] is treated as a candidate leaf action that must outperform all physical actions under the Evaluator’s utility score  ($U(\mathcal{S},a)$). This explicit utility-based competition serves as a gating mechanism, executing communication only when its expected information gain justifies the cost relative to physical exploration.
>
>  As suggested, we have revised Section 4.3 (Composer) to state *"Specifically, each leaf node is assigned an action aimed at handling the uncertainty of the given path—either a physical action for direct interaction or a communication action for sharing information or instructing the collaborator."* ensuring this mechanism is clear to all readers.

---

> > ### Author Response · Authors · 2025-11-26
> >
> > > W2 & Q1: Composer's Reliability
> >
> > We agree that the reliability of the Composer is central to the framework. To directly address this concern, we performed two complementary analyses that measure its robustness, hit rate, and behavior under vague or ambiguous reasoning traces.
> >
> > **(1) Quantitative Stress Test at the Assumption Level.**
> >
> > To directly address the reviewer’s concern about whether the Composer reliably identifies critical uncertainties, especially when the Planner trace is vague, we conducted a quantitative stress test measuring its assumption-level accuracy and downstream robustness. Since the correctness of extracted or generated assumptions is inherently semantic and context-dependent, it cannot be reliably validated by rule-based metrics or environment rewards alone. Human experts are therefore required to serve as the only trustworthy reference for judging whether an assumption genuinely reflects the Planner’s intent or constitutes a reasonable hypothesis in ambiguous contexts.
> >
> > This experiment is framed as a stress test because it deliberately includes reasoning traces that are vague, underspecified, or noisy, precisely the situations where LLM-based systems are most likely to hallucinate or misinterpret implicit intent. By measuring assumption-level accuracy across increasing levels of ambiguity, we directly evaluate whether the Composer remains robust when its input deviates from ideal conditions.
> >
> > **[1] Experimental Design.**
> >
> > Using execution logs from 10 C-WAH tasks, we evaluated the Composer at the level of individual assumption nodes. We modified the prompt so that each assumption explicitly self-labels whether it is extracted directly from the Planner’s reasoning trace or generated as a new hypothesis not explicitly stated in the trace.
> >
> > To stress-test the Composer under varying levels of ambiguity, four domain experts categorized each instance into easy, medium, or hard based on trace clarity. Easy cases contain assumptions explicitly stated with minimal noise. Medium cases involve assumptions that are inferable but embedded in ambiguous or cluttered reasoning. Hard cases correspond to vague traces where the critical assumption is not directly mentioned, requiring implicit inference or hypothesis generation.
> >
> > Each assumption was then independently evaluated by three additional experts to provide an objective reference for correctness. Extracted assumptions were considered valid only if explicitly supported by the trace, while generated assumptions were considered valid if they formed a contextually plausible hypothesis given the environment, goal, and agent memory. Final labels were determined by majority vote, and the Composer’s hit rate was defined as the proportion of assumptions judged valid, reported separately across difficulty levels.
> >
> > Beyond assumption-level accuracy, we further examined system-level robustness by analyzing how the Evaluator handles imperfect assumptions. We compared scenarios composed entirely of valid assumptions with scenarios containing at least one invalid assumption and measured their Likelihood and Utility scores. This design explicitly tests whether erroneous assumptions meaningfully propagate into action selection or are suppressed by the Evaluator’s scoring mechanism.
> >
> > **[2] Results: Composer Hit Rate.**
> >
> > | **Easy** | **Medium** | **Hard** | **Overall** |
> > | --- | --- | --- | --- |
> > | 84.3% | 77.8% | 76.7% | 80.6% |
> >
> > The Composer achieves 80.6% overall validity and maintains robust performance even in highly ambiguous traces, demonstrating that it reliably identifies meaningful uncertainties rather than hallucinating arbitrary ones.
> >
> > **[3] Results: Robustness to Invalid Assumptions**
> >
> > To verify that occasional incorrect assumptions do not destabilize the system, we analyzed how the Evaluator reacts to such cases. Scenarios containing invalid assumptions received substantially lower Likelihood and Utility scores than fully valid scenarios.
> >
> > | **Scenario Type** | **Likelihood** | **Utility Score** |
> > | --- | --- | --- |
> > | **Valid assumptions only** | 3.28 | 3.63 |
> > | **Contains invalid assumptions** | 2.85 | 2.51 |
> >
> > Scenarios containing invalid assumptions show a clear degradation in both Likelihood and Utility. While the Likelihood drop (3.28 → 2.85) reflects the Evaluator’s ability to detect epistemic inconsistency, the larger drop in Utility (3.63 → 2.51) is more critical: it indicates that flawed assumptions are not merely identified but actively prevented from dominating action selection.
> >
> > This demonstrates that PCE is not vulnerable to occasional Composer errors. The Evaluator serves as a probabilistic safety layer that attenuates the influence of incorrect assumptions by translating them into lower expected utility, thereby preserving decision stability even when assumption extraction is imperfect.

---

> > > ### Author Response · Authors · 2025-11-26
> > >
> > > **(2) Expert Evaluation at the Decision-Tree Level.**
> > >
> > > While assumption-level accuracy verifies whether individual nodes are valid, it does not guarantee that these nodes are composed into a coherent and decision-ready structure. In multi-agent planning, even locally correct assumptions can lead to unstable behavior if their global organization is inconsistent, poorly ranked, or misaligned with actions. We therefore conducted a complementary evaluation at the decision-tree level to assess whether the Composer produces trees that are structurally sound and logically coherent.
> > >
> > > We conducted a qualitative comparison between Composer-generated decision trees and human-expert-generated decision trees.
> > >
> > > **[1] Experimental Setup**
> > >
> > > Using the execution logs from the previous experiment, four domain experts manually constructed a *Decision Trees Dataset* based on the identical input prompts provided to the Composer.  Both the Composer-generated trees and the expert-generated trees (excluding their own) were then jointly evaluated through blind cross-review on a 7-point Likert scale across five dimensions.
> > >
> > > **[2] Results and Interpretation**
> > >
> > > | **Metric** | **Composer** | **Human Expert** | **Evaluation Focus** |
> > > | --- | --- | --- | --- |
> > > | **Q1: Extraction Accuracy** | 6.27 | 6.64 | Does the extracted assumption actually exist in the trace? |
> > > | **Q2: Generation Capability** | 6.39 | 6.73 | Are new assumptions generated appropriately when the trace is vague? |
> > > | **Q3: Ranking Logic** | 5.83 | 6.23 | Are nodes ordered by uncertainty reduction & impact? |
> > > | **Q4: Logical Consistency** | 6.18 | 6.55 | Are there contradictions in the decision-tree? |
> > > | **Q5: Action Appropriateness** | 5.98 | 6.15 | Do leaf actions match the scenario path? |
> > >
> > > Overall, the Composer-generated decision trees achieve near–expert-level quality across all five dimensions. The strong scores in *Generation Capability* and *Logical Consistency* indicate that the Composer not only extracts assumptions accurately but also generates new assumptions appropriately when needed, while maintaining coherence with previously stated assumptions. The results for *Ranking Logic* show that the Composer can position critical assumptions in a meaningful order that effectively reduces uncertainty. Finally, the *Action Appropriateness* score suggests that the Composer assigns suitable leaf actions that correctly correspond to the scenario paths represented in the decision tree.
> > >
> > > **(3) Conclusion.**
> > >
> > > We conducted two additional evaluations at both the assumption level and the decision-tree level to assess the Composer’s reliability. These experiments demonstrate that even when the input reasoning trace is ambiguous, the Composer maintains high accuracy in extracting and generating assumptions, and ultimately produces useful decision trees built upon critical, non-contradictory assumptions with appropriately assigned actions. We include the full details of these results in Appendix A.11.

---

> > > > ### Author Response · Authors · 2025-11-27
> > > >
> > > > > Response to Ethics Concern Raised by Reviewer
> > > >
> > > > We acknowledge the ethics flag raised regarding our human subject experiments and clarify the scope and ethical handling of our user studies.
> > > >
> > > > Our studies consist of a small-scale user study conducted entirely within simulated environments and an expert evaluation performed through technical analysis of model-generated outputs. In the user study, twelve participants interacted with an embodied agent in a controlled virtual task environment, and the expert evaluation involved examining the validity of system-generated decision trees and assumptions from a technical standpoint. Across both components, no personal, sensitive, behavioral, or intervention-related data was collected, and all interactions were limited to task-level observations or analytical judgments of model outputs.
> > > >
> > > > Participation in both studies was entirely voluntary, and no personally identifying information was collected or recorded at any stage. All contributors were informed that their responses would be used exclusively for academic research purposes.
> > > >
> > > > We believe this design aligns with responsible research practices for low-risk user and expert evaluation studies commonly conducted in AI system assessment.

---

### Official Review · Reviewer_5pEK · 2025-11-01

**Soundness:** 2
**Presentation:** 3
**Contribution:** 2
**Rating:** 4
**Confidence:** 4

**Summary:**

The paper proposes PCE (Planner–Composer–Evaluator) for embodied decision making. PCE is a modular framework that turns implicit assumptions inside an LLM’s reasoning trace into a structured decision-tree for uncertainty-aware planning in partially observable multi-agent environments. Experiments (including human evaluations) on C-WAH and TDW-MAT show that PCE consistently outperforms communication-centric baselines  across three LLMs. This work also provides ablation and user studies.

**Strengths:**

1. It is interesting to introduce an uncertainty-handling mechanism to this field. PCE explicitly extracts and evaluates the LLM’s latent assumptions and plan with structured decision-tree.
2. The experimental results show strong gains in success rate and step efficiency on two benchmarks, with comparable computational cost. It is also good to include human evaluations.
3. The ablations on reasoning and different LLMs also show the consistency of performance gain.

**Weaknesses:**

1. It is unclear how the hyperparameters were chosen. The authors set D = 3, alpha = 1, beta = 1, lambda = 1, Kaction = 10, Kmessage= 3 empirically, but no according further explaination or ablation is provided.
2. The related work section should discuss tree-search-based methods (e.g., CoTS) more clearly.  The authors need to clearly articulate how their method differs conceptually and why PCE is needed beyond existing tree reasoning or search frameworks.
3. The paper would benefit from more case studies or qualitative analyses to illustrate how PCE behaves in different uncertainty scenarios and to provide deeper insight into its decision-making process.

**Questions:**

See weaknesses, and

1. Why is the Usage of PCE lower than CoELA? According to my understanding, PCE’s three modules all require LLM inference, which should make the total cost higher than CoELA, which only infers twice. Please clarify this discrepancy.
2. How about using MCTS based on distance as a comparison or baseline? It might provide a stronger reference for tree-based planning efficiency.

I am willing to change my score if the concerns are addressed

---

> ### Author Response · Authors · 2025-11-24
>
> Thank you for the thoughtful and constructive feedback. Below, we provide our point-by-point responses and hope they satisfactorily address your concerns. We look forward to further discussion that may help refine the work.
>
> > W1: Hyperparameter Choices and Ablation Analysis
>
> We agree that empirical justification is essential, given their influence on planning depth, communication frequency, and execution efficiency.
>
> To validate our choices, we conducted a sensitivity analysis on C-WAH, varying each parameter while holding others fixed.  The table below reports the performance (Total Steps) across variations of key hyperparameters: tree depth ($D=3$), cost coefficients ($\alpha=1, \beta=1$), penalty weight ($\lambda=1$), and memory history lengths ($K_{action}=10, K_{message}=3$). Our default setting achieved 42.76 steps.
>
> **Table: Hyperparameter sensitivity analysis results**
>
> | Hyperparameter | Variant 1 | Total Steps | Variant 2 | Total Steps |
> |----------------|-----------|-------------|-----------|-------------|
> | **Tree Max Depth ($D$)** | $D$ = 2 | 44.6 | $D$ = 4 | 42.4 |
> | **Cost Weight (α)** | α = 0.5 | 50.1 | α = 1.5 | 45.5 |
> | **Cost Weight (β)** | β = 0.5 | 48.6 | β = 1.5 | 44.6 |
> | **Global Penalty (λ)** | λ = 0.5 | 58.7 | λ = 1.5 | 45.4 |
> | **Memory Lengths ($K_{action}$)** | $K_{action}$ = 5 | 44.4 | $K_{action}$ = 15 | 44.3 |
> | **Memory Lengths ($K_{message}$)** | $K_{message}$ = 2 | 49.5 | $K_{message}$ = 4 | 49.7 |
>
> **Tree Max Depth D.** Performance stabilizes around $D = 3$. Reducing D limits the ability to explore diverse assumption scenarios, while increasing D yields marginal gains but introduces exponential tree growth in the worst case, which amplifies hallucination risk and computational cost. This supports  $D = 3$ as a reasonable trade-off.
>
> **Cost weights $\alpha, \beta$.** Lower $\alpha$ or higher $\beta$ underestimates physical action cost, leading to inefficient over-exploration; conversely, higher $\alpha$ or lower $\beta$ biases the agent toward excessive communication. Both degrade overall efficiency.
>
> **Global penalty $\lambda$.** Lower $\lambda$ overemphasizes likelihood and gain, while higher $\lambda$ leads to overly conservative, short-sighted decisions, explaining the observed performance drop.
>
> **Memory lengths $K_{action}, K_{message}$** . Shorter memory weakens long-horizon planning and coordination, while longer memory introduces irrelevant history and inflates input context. Sensitivity is particularly high for $K_{message}$ due to its critical role in partial-observable multi-agent coordination.
>
> Overall, these results confirm that our defaults lie within a stable operating region that balances reasoning coverage, cost awareness, and memory relevance, and are not arbitrarily chosen.
> We have included these results in Appendix A.5 of the revised paper.
>
> > W2: Why PCE Is Conceptually Necessary Beyond Existing Tree-Based Frameworks
>
> We revised the Related Work section to clearly articulate the conceptual gap between PCE and prior tree-based approaches such as Tree of Thoughts (ToT) and CoTS.
>
> The key distinction lies not in how a tree is searched, but in what the tree represents:
>
> 1. ToT constructs a tree over *reasoning steps* to enhance logical coherence, but it operates primarily within the internal reasoning space of a single agent. It implicitly assumes a fully observable environment and treats tree nodes as cognitive steps rather than probabilistic environmental states, limiting its applicability in dynamic, partially observable multi-agent scenarios.
> 2. CoTS performs collaborative tree search over the joint-reasoning-and-action space. However, it relies on communication-driven exploration to generate joint space and prune candidate actions, making dialogue a prerequisite for planning and introducing significant latency and token overhead.
> 3. **PCE (ours)**, in contrast, reframe the tree as an explicit representation of environmental uncertainty, where internal nodes encode assumptions about the world and leaves correspond to candidate actions. Crucially, communication is not the mechanism of search, but one selectable action among others, evaluated by expected utility. This structural shift enables principled trade-offs between physical exploration and communication, resolving the efficiency bottleneck inherent in communication-centric tree search.

---

> ### Author Response · Authors · 2025-11-24
>
> > W3: Illustrating How PCE Corrects Uncertainty-Induced Planning Failures
>
> We agree that qualitative analysis is essential for understanding PCE’s decision-making process. We have added a new section, “Appendix A.7: Qualitative Case Studies on Uncertainty Handling,” presenting multiple scenarios that reveal how PCE revises suboptimal plans arising from ambiguous or misleading reasoning traces.
>
> Representative Example: Correcting Misguided Spatial Exploration
>
> - **Scenario:** Agent Alice must find the last target item (loaf_bread). She is currently in the <'Livingroom'>. The <'Bedroom'> has been confirmed empty, while the <'Kitchen'> and <'Office'> remain unexplored.
> - **Planner Failure:** The Planner mistakenly proposes revisiting the <'Bedroom'>, failing to incorporate negative search history.
> - **PCE Process:**
>     1. **Composer** extracts the core uncertainty *"Where is the remaining loaf_bread?"* and constructs a scenario tree branching into:
>         - **Assumption A:** "Bread is in the <'Bedroom'>" (reason: Previous check was inaccurate).
>         - **Assumption B:** "Bread is in the <'Kitchen'>" (reason: Item is in an unexplored area).
>     2. **Evaluator** scores “Scenario with Assumption A” with extremely low Likelihood (due to search history) and “Scenario with Assumption B” with high Likelihood.
> - **Outcome:** PCE overrides the Planner and select [goexplore] <'Kitchen'>.
>
> This example demonstrates how PCE effectively filters out “hallucinated” or inefficient plans through structured evaluation. In the revised appendix, we also include other cases, illustrating PCE's versatility.
>
> > Q1: Why PCE Achieves Lower LLM Usage Despite More Modules
>
> We clarify that Usage measures total LLM token consumption per episode, not per-step inference cost. While PCE incurs higher cost per step due to its three-module design, it substantially reduces the number of steps required to complete a task.
>
> In C-WAH benchmark, PCE completes episodes in 42.76 steps, whereas CoELA requires 60.40. This ~30% reduction in episode length dominates the additional module overhead, resulting in lower total token usage.
>
> Furthermore, due to the absence of an explicit cost-evaluation mechanism, CoELA implicitly defaults to treating communication as a low-cost action, which results in frequent dialogue loops (CoELA: 9.88 comms vs. PCE: 1.70 comms). Each loop triggers an additional planning cycle, amplifying token consumption. In contrast, PCE explicitly evaluates communication utility and invokes it only when it yields higher expected value than physical action, effectively suppressing redundant communication-planning cycles.
>
> Since every communication action consumes a simulation step and triggers a subsequent re-planning (LLM inference) cycle, Baseline approaches inefficiently use tokens for redundant interactions. PCE’s Evaluator scores communication utility against physical actions, executing it only when essential, thereby pruning these inefficient "communication-planning loops."
>
> Therefore, PCE’s lower Usage is a direct consequence of its efficiency-oriented design: modular reasoning minimizes unnecessary steps, which reduces total LLM cost.
>
> > Q2: Incorporating MCTS-based method as an Additional Baseline
>
> Following the reviewer’s suggestion, we incorporated the MCTS-based Hierarchical Planner (MHP) from Watch-And-Help challenge [1] as an additional baseline. MHP utilizes MCTS to find optimal action trajectories by minimizing graph traversal costs and step counts (e.g., optimizing based on topological connectivity and action costs).
>
> **Comparative Results (C-WAH, Total Steps):**
>
> - PCE (Ours): 42.76
> - MHP (MCTS-based) [1]: 64.90
>
> **Analysis:**
> While MHP effectively optimizes path efficiency, it operates purely on distance heuristics and does not reason about latent semantic uncertainty in partially observable environments. In contrast, PCE explicitly models alternative hypothesis spaces (inferring “the apple is likely in the kitchen” from context rather than physically searching rooms based on graph proximity) and prioritizes actions based on expected utility rather than proximity alone.
>
> These results demonstrate that distance-based MCTS, although strong for deterministic navigation, is structurally insufficient for embodied POMDP settings where resolving semantic uncertainty (not merely minimizing movement cost) is the primary bottleneck in effective planning.
>
> The results have been added to Appendix A.8 (Additional Baselines) in the revised manuscript
>
> ---
>
> References: [1] Puig et al., “Watch-and-help: A challenge for social perception and human-ai collaboration,” ICLR 2021.

---

> > ### Comment · Reviewer_5pEK · 2025-11-26
> > **Concerns Addressed**
> >
> > Thank you for your response. I have read the content carefully and most of my concerns are addressed. I will raise my score to 6. I would also appreciate if you put the discussion to Q1 in the paper.

---

> > > ### Author Response · Authors · 2025-11-27
> > >
> > > Thank you for raising your score. We sincerely appreciate the time you invested and your constructive evaluation.
> > >
> > > We have incorporated the discussion related to Q1 into Section 5.1 (Comparative Results) of the manuscript. Let us know if any additional clarification is required.

---

### Meta-Review · Area_Chair_KaF6 · 2026-01-06

**Summary:**

Summary:
This paper introduces PCE (Planner–Composer–Evaluator), a modular framework for uncertainty-aware decision making in partially observable, multi-agent embodied environments. Evaluations on the C-WAH and TDW-MAT benchmarks across multiple LLM backbones show that PCE consistently improves task success and efficiency while reducing unnecessary communications.


Strengths:
1. The uncertainty-handling mechanism is interesting.
2. The method has a strong performance on two benchmarks without introducing higher computational costs.
3. The ablation studies show the clear benefit of the method.


Weaknesses:
1. It is unclear how the hyperparameters were chosen.
2. The related work section should discuss tree-search-based methods (e.g., CoTS) more clearly.
3. Lack of discussion on prior work on long-horizon LLM planning
4. The paper would benefit from more case studies or qualitative analyses.
5. It is unclear whether the method can scale to larger numbers of agents.
6. The paper offers no analysis of the Composer's reliability.
7. The paper's claim that PCE "amplifies the benefits of scaling" seems slightly overstated; the data suggest the benefit of the PCE framework is largely additive—PCE starts at a better baseline, and that baseline advantage is maintained or slightly widened as the model scales.
8. More discussions on the difference between the existing methods and PCE.
9. PCE relies on many existing methods.
10. The analysis lacks clarity on why performance exceeds baselines even without the Composer, and where specifically the Composer contributes to performance improvements.
11. The components in PCE rely on LLMs, which may suffer from stability issues caused by LLMs.
12. It is unclear whether the system is sending too many tokens at one time in the communication channel or if the system has a higher latency in planning for the next step.

**Reviewer Concerns:**

I think all reviewer concerns have been addressed. The authors have either provided a revision to clarify relevant points or included new results/analyses to support their claims.

**Reviewer Scores:**

Reviewer 5pEK may raise their score as their concerns have been addressed, in my opinion.

Reviewer FyaB may raise their score as the authors have adequately addressed their concerns with the new experiments and the revision.

Reviewer SbnL may raise their score, as I also find the rebuttal has adequately addressed the concerns.

Reviewer yuvs has already indicated that they would maintain their initial score in their response.

In sum, I believe the reviewers who had negative scores will likely raise their score marginally, while the reviewer who gave a positive score has indicated that they would maintain their initial rating.

---

### Decision · Program_Chairs · 2026-01-26

Accept (Poster)